# YAHPO Gym - An Efficient Multi-Objective Multi-Fidelity Benchmark for Hyperparameter Optimization

**Florian Pfisterer**[1,2] **Lennart Schneider**[1,2] **Julia Moosbauer**[1] **Martin Binder**[1] **Bernd Bischl**[1]

[1]Department of Statistics, LMU Munich, Germany
[2]Equal contributions

**Abstract** When developing and analyzing new hyperparameter optimization methods, it is vital to empirically evaluate and compare them on well-curated benchmark suites. In this work, we propose a new set of challenging and relevant benchmark problems motivated by desirable properties and requirements for such benchmarks. Our new surrogate-based benchmark collection consists of 14 scenarios that in total constitute over 700 multi-fidelity hyperparameter optimization problems, which all enable multi-objective hyperparameter optimization. Furthermore, we empirically compare surrogate-based benchmarks to the more widely-used tabular benchmarks, and demonstrate that the latter may produce unfaithful results regarding the performance ranking of HPO methods. We examine and compare our benchmark collection with respect to defined requirements and propose a single-objective as well as a multi-objective benchmark suite on which we compare 7 single-objective and 7 multi-objective optimizers in a benchmark experiment. Our software is available at [https://github.com/slds-lmu/yahpo_gym].

## 1 Introduction

Hyperparameter optimization (HPO) of machine learning (ML) models is a crucial step for achieving good predictive performance [43]. Over the last ten years, a large and still growing set of HPO tuning methods based on different principles has been developed [31, 66, 38]. A particularly interesting development are multi-fidelity methods, which make use of relatively cheap approximations of a given true objective, thereby achieving good performance relatively quickly [44, 21, 35], as well as multi-objective methods, which allow for simultaneous optimization of multiple objectives [40]. While different HPO methods found considerable adoption in practice, it is by no means clear which method performs best under which circumstances. In order to investigate this, it is necessary to evaluate these methods on testbeds that are ideally *i*) highly efficient, *ii*) include a sufficient amount of representative and diverse benchmark instances and *iii*) are easy to set up and integrate with different optimizer APIs. Furthermore, benchmarks have found use in *meta-learning* [70, 74, 59] and *meta-optimization* [49, 53]. In those settings, a larger number of potentially relevant optimization problems is required in order to obtain results that generalize beyond the set of (meta-)training instances. Simultaneously, those applications require a large number of evaluations that make obtaining real evaluations prohibitively expensive, indicating a need for benchmarks that are cheap to query.

Several benchmarks that aim to address this, each of which are collections of multiple benchmark instances, have been proposed [69, 15, 60, 19]. Benchmark instances can be classified into four categories: (i) synthetic functions, (ii) benchmarks incorporating *real* evaluations, (iii) *tabular* benchmarks based on pre-evaluated grid points, and (iv) *surrogate* benchmarks making use of meta-models that approximate the relationship between configurations and performance metrics. Each category has various advantages and drawbacks. Synthetic functions can be evaluated quickly but are often not representative for the type of problems encountered in practice; real evaluations on the other hand are often prohibitively expensive, especially in the context of larger

benchmarks and neural architecture search (NAS). Tabular benchmarks, while cheap to evaluate, rely on a pre-defined grid which changes the optimization problem and can potentially lead to biases. Surrogate benchmarks are also cheap to query but require high quality surrogates in order to avoid introducing bias. While benchmark suites have found some use in scientific publications, they are not used ubiquitously. This lack of permeation – and consequently the lack of a standard test bed – can result in researchers choosing benchmark problems that favor their own method, leading to the publication of biased results. The problem of *cherry picking*, also termed *rigging the lottery* [14], can be ameliorated through the use of standardized testing infrastructure along with a detailed definition of evaluation criteria that are widely adapted.

We therefore observe a clear need for benchmark libraries that provide unified interfaces to a variety of cheap to evaluate, realistic, and practically relevant benchmarking problems that are defined across diverse search spaces. In this work, we propose *YAHPO Gym*, a *surrogate-based* benchmark library including a collection of over 700 benchmark instances defined across 14 *scenarios*. Scenarios are comprised of evaluations of one given machine learning algorithm on different datasets (= instances) and therefore share the same search space and performance metrics. It contains a versioned set of surrogate models that allow for *multi-fidelity* evaluations of *multiple objectives*. Our library is licensed under the *Apache 2.0* license and can be freely used and extended by the community. Usage and available functionality is extensively documented[1].

**Contributions**: We introduce YAHPO Gym, a surrogate-based benchmark for machine-learning HPO. We conceptually demonstrate that tabular benchmarks may induce bias in performance estimation and ranking of HPO methods, and that this happens to a lesser degree with surrogate benchmarks. We argue that our surrogate benchmark YAHPO Gym meets all desiderata for a good benchmark, providing faithful results, fast evaluation, relevant problems and realistic objective landscapes both on local as well as global scales. In order to demonstrate this, we conduct an extensive evaluation of the proposed surrogates indicating that our surrogate models indeed provide high quality approximations. We propose two benchmark suites for *single-objective* and *multi-objective* evaluation comprised of a subset of our instances and demonstrate how they can be used with YAHPO Gym in a *multi-fidelity* and a *multi-objective* optimization benchmark.

## 2 Related Work

Several efforts to provide unified testbeds for black-box optimization exist. For general purpose black-box optimization, COCO [29] provides a collection of various synthetic black-box benchmark functions, while *kurobako* [56] is a collection of various general black-box optimizers and benchmark problems. Similarly, *Bayesmark* [69] includes several benchmarks for Bayesian Optimization on real problems and *LassoBench* [64] provides a benchmark for high-dimensional optimization problems. *HPOlib* [15] was one of the first to propose a common test bed for empirically assessing the performance of HPO methods. It provides a common API to access synthetic test functions, real-world HPO problems, tabular benchmarks as well as some surrogate benchmarks and found use in empirical benchmark studies [6]. Its successor *HPOBench* [19] offers similar capabilities, focussing on reproducible containerized benchmarks. It offers 12 benchmark scenarios and more than 100 test instances. Recently, [60] introduced *HPO-B*, a large-scale reproducible (tabular) benchmark for black-box HPO based on OpenML [71]. *HPO-B*[2] relies on 16 search spaces that were evaluated sparsely on 101 datasets. *PROFET* [37] in contrast is not based on real datasets but uses a generative meta-model to generate synthetic but realistic benchmark instances. In the past, tabular benchmarks have been used frequently to speed up experiments in the context of HPO [66, 23, 72, 22] and NAS (c.f. [50]). Eggensperger et al. [17] compared

---

[1]Documentation and data are available at `https://github.com/slds-lmu/yahpo_gym`.
[2]We consider the published v2 version for comparison. Surrogates are only available in the v3 version.

Table 1: Comparison of HPO Benchmark Suites.

| Suite | Types | #Collections | #HPs | MF | MO | TF | Async | H | Time$^\dagger$ | Memory$^\dagger$ |
|-------|-------|-------------|------|----|----|----|-------|---|------|--------|
| YAHPO Gym | S | 14 | 2-38 | ✓ | ✓ | ✓ | (-) | ✓ | $0.4^*s$ | 0.1 GB |
| HPOBench | R/T/S | 12 | 4-26 | ✓ | ✓ | (-) | − | (-) | 12.2s | 0.2 GB |
| HPO-B (v2) | T/(S) | 16 | 2-18 | − | − | ✓ | − | − | 18.8s | 3.7 GB |

MF: Multi-fidelity; MO: Multi-objective, TF: Transfer-HPO, Async: Asynchronous evaluation; H: hierarchical search spaces.
✓: fully supported; (-): partially supported; -: not supported; R/T/S:real/tabular/surrogate.
$^\dagger$: Runtime and memory footprint for 300 iterations of Random Search on an SVM instance. *: allowing for batched evaluation, YAHPO Gym takes only $0.13s$.

different instance surrogate models for 9 different HPO problems and concluded that the results of benchmarks run on surrogate models generally closely mimic those of benchmarks using the actual evaluations that they are derived from, if performance measures of the surrogate models indicate that they predict the underlying objective values sufficiently well (cross-validated Spearman's $\rho$ between 0.9 and 1 [17]). Similar observations have been made in the context of algorithm configuration [18] and NAS [65].

We compare YAHPO Gym with the recently published benchmarks HPOBench [19] and HPO-B [60] in Table 1. Our library relies on high quality surrogates that allow for *multi-fidelity* as well as *multi-objective* evaluation. While existing benchmark suites could in principle be used to construct multi-objective benchmarks, they do not offer full support: HPOBench contains only few instances that allow evaluating multiple metrics and offers no unified API to query those, while HPO-B does not support multiple objectives at all. Furthermore, neither propose a concrete evaluation protocol, opening up a multiplicity of (benchmark) design choices which can lead to inconclusive results (c.f. [55]). Instead of relying on *containerization* to allow for portability, our library relies on neural network surrogates compressed using *ONNX* [3], allowing for reproducibility and portability while simultaneously being extremely fast and efficient due to minimal overhead. This is demonstrated in a small experiment where we measure runtime and memory consumption for evaluating 300 random configurations on SVM search spaces also shown in Table 1, demonstrating that our software is more time and memory efficient. See details in Supplement B.2. While YAHPO Gym provides the flexibility to design and execute any subset of the provided benchmarks, we also propose two fully specified testbeds for single- and multi-objective optimization that were specifically selected to cover a diverse set of relevant instances while being less extensive. See details in Supplement E.2 and Supplement E.3.

## 3 Background

### 3.1 Hyperparameter Optimization

An ML *learner* or *inducer* $\mathcal{I}$ configured by hyperparameters $\boldsymbol{\lambda} \in \Lambda$ maps a dataset $\mathcal{D} \in \mathbb{D}$ to a model $\hat{f}$, i.e., $\mathcal{I} : \mathbb{D} \times \Lambda \rightarrow \mathcal{H}, (\mathcal{D}, \boldsymbol{\lambda}) \mapsto \hat{f}$. HPO methods for ML aim to identify a well-performing hyperparameter configuration (HPC) $\boldsymbol{\lambda} \in \tilde{\Lambda}$ for $\mathcal{I}_{\boldsymbol{\lambda}}$ [10]. Typically, the considered search space $\tilde{\Lambda} \subset \Lambda$ is a subspace of the set of all possible HPCs: $\tilde{\Lambda} = \tilde{\Lambda}_1 \times \tilde{\Lambda}_2 \times \cdots \times \tilde{\Lambda}_d$, where $\tilde{\Lambda}_i$ is a bounded subset of the domain of the $i$-th hyperparameter $\Lambda_i$. This $\tilde{\Lambda}_i$ can be either real, integer, or category valued, and the search space can contain dependent hyperparameters, leading to a possibly hierarchical search space. We formally define the (potentially multi-objective) HPO problem as:

$$\boldsymbol{\lambda}^* \in \underset{\boldsymbol{\lambda} \in \tilde{\Lambda}}{\arg\min}\, c(\boldsymbol{\lambda}), \quad \text{with} \quad c : \tilde{\Lambda} \rightarrow \mathbb{R}^m, \tag{1}$$

where $\boldsymbol{\lambda}^*$ denotes the theoretical optimum and $c$ maps an arbitrary HPC to (possibly multiple) target metrics. The classical HPO problem is defined as $\boldsymbol{\lambda}^* \in \arg\min_{\boldsymbol{\lambda} \in \tilde{\Lambda}} \widehat{\mathrm{GE}}(\boldsymbol{\lambda})$, i.e., the goal is

to minimize the estimated generalization error, see [10] for further details. Instead of optimizing only for predictive performance, other metrics such as model sparsity or computational efficiency of prediction (e.g., MACs and FLOPs or model size and memory usage) could be included, resulting in a multi-objective HPO problem [62, 30, 7, 57, 27]. $c(\boldsymbol{\lambda})$ is a black-box function, as it usually has no closed-form mathematical representation, and analytic gradient information is generally not available. Furthermore, the evaluation of $c(\boldsymbol{\lambda})$ can take a significant amount of time. Therefore, the minimization of $c(\boldsymbol{\lambda})$ forms an *expensive black-box* optimization problem.

Many HPO problems allow for approximations of the objective to a varying fidelity, making *multi-fidelity optimization* a viable option [44, 62, 35]. For example, in the context of fitting neural networks, it is possible to stop or pause training runs early when performance does not indicate a promising final result [67]. Another possibility is given by reducing the fraction of the dataset $\mathcal{D}_{\text{train}}$ used for training [38], since the complexity of evaluating $c(\boldsymbol{\lambda})$ is often at least linear in $|\mathcal{D}_{\text{train}}|$. Formally, the possibility of multi-fidelity evaluation can be represented in the form of a "budget" hyperparameter which we denote by $\lambda_{\text{budget}}$ as a component of $\boldsymbol{\lambda}$.

### 3.2 Hyperparameter Optimization Benchmarks

Benchmark suites are comprised of a set of benchmark *instances* that each define an optimization problem to be solved. We formally define benchmark instances adapted from [19] as:

**Definition 1 (Benchmark Instance)** *A benchmark instance consists of a function $g : \Lambda \to \mathbb{R}^m, m \in \mathbb{N}^+$, and a bounded hyperparameter space $\tilde{\Lambda}$ which is the Cartesian product of hyperparameters $\tilde{\Lambda}_1, \ldots, \tilde{\Lambda}_d$. Multi-fidelity benchmarks can be queried at lower fidelities by varying the budget parameter $\tilde{\Lambda}_{budget} \in \tilde{\Lambda}$. While hyperparameters $\tilde{\Lambda}_i$ can be continuous, integer, ordinal or categorical, we require at least ordinal scales for the fidelity parameter(s) $\Lambda_{budget}$. We call a benchmark instance multi-objective if the number of objectives $m > 1$ and single-objective otherwise.*

We consider HPO benchmark instances estimating the generalization error $g(\boldsymbol{\lambda}) = \widehat{\text{GE}}(\mathcal{I}, \mathcal{J}, \rho, \boldsymbol{\lambda})$ given an inducer $\mathcal{I}$, resampling $\mathcal{J}$, and performance metric(s) $\rho$, along with other possibly relevant metrics (computational cost, memory, ...). *Real* instances are based on actually performing these evaluations during the benchmark, while *tabular* instances are based on a fixed set of pre-recorded evaluations. Instances based on *surrogates* in turn approximate the functional relationship between $\boldsymbol{\lambda}$ and $g(\boldsymbol{\lambda})$. For clarity, we provide more precise definitions of *synthetic, tabular* and *surrogate* instances in Supplement B.3. *Real* instances rely on live evaluations of the generalization error and are therefore often prohibitively computationally expensive, especially when considering larger benchmarks or meta-learning scenarios across many tasks [70, 59, 24]. Practitioners therefore often rely on *tabular* or *surrogate* benchmarks for large benchmark studies because they are often cheaper to evaluate by orders of magnitude. For *tabular* benchmarks, a large collection of pre-computed hyperparameter performance mappings is provided, which serves as a look-up table during runs of HPO methods. This has the downside of constraining the search space to precomputed evaluations, essentially turning the optimization problem from a *continuous/mixed space* to a *discrete* optimization problem. *Surrogate* benchmarks can strike a balance between the efficiency and faithful approximation to the real problem by learning the functional relationship between hyperparameters and performance values yielding an approximation $\hat{g}(\boldsymbol{\lambda})$ of $g(\boldsymbol{\lambda})$. This allows evaluations across the full search space $\tilde{\Lambda}$ while being considerably cheaper to evaluate. The usefulness of surrogates in turn relies on the approximation quality of the surrogate model. We present an in-depth analysis of approximation qualities of the surrogates employed in YAHPO Gym in Supplement E.1.

**Definition 2 (Benchmark Scenario)** *A benchmark scenario consists of a set of $K$ functions $g_k : \Lambda \to \mathcal{Y} \subseteq \mathbb{R}^m, m \in \mathbb{N}^+, k \in \{1, ..., K\}$ corresponding to a set of Benchmark Instances. Each instance within a scenario shares the same bounded hyperparameter space $\tilde{\Lambda}$ (and therefore fidelity parameters) as well as the same co-domain $\mathcal{Y}$.*

Table 2: YAHPO Gym Benchmarks.

| Scenario | Search Space | #Instances | Target Metrics | Fidelity | H |
|---|---|---|---|---|---|
| rbv2_super | 38D: Mixed | 103 | 9: perf(6) + rt(2) + mem | fraction | ✓ |
| rbv2_svm | 6D: Mixed | 106 | 9: perf(6) + rt(2) + mem | fraction | ✓ |
| rbv2_rpart | 5D: Mixed | 117 | 9: perf(6) + rt(2) + mem | fraction | |
| rbv2_aknn | 6D: Mixed | 118 | 9: perf(6) + rt(2) + mem | fraction | |
| rbv2_glmnet | 3D: Mixed | 115 | 9: perf(6) + rt(2) + mem | fraction | |
| rbv2_ranger | 8D: Mixed | 119 | 9: perf(6) + rt(2) + mem | fraction | ✓ |
| rbv2_xgboost | 14D: Mixed | 119 | 9: perf(6) + rt(2) + mem | fraction | ✓ |
| nb301 | 34D: Categorical | 1 | 2: perf(1) + rt(1) | epoch | ✓ |
| lcbench | 7D: Numeric | 34 | 6: perf(5) + rt(1) | epoch | |
| iaml_super | 28D: Mixed | 4 | 12: perf(4) + inp(3) + rt(2) + mem(3) | fraction | ✓ |
| iaml_rpart | 4D: Numeric | 4 | 12: perf(4) + inp(3) + rt(2) + mem(3) | fraction | |
| iaml_glmnet | 2D: Numeric | 4 | 12: perf(4) + inp(3) + rt(2) + mem(3) | fraction | |
| iaml_ranger | 8D: Mixed | 4 | 12: perf(4) + inp(3) + rt(2) + mem(3) | fraction | ✓ |
| iaml_xgboost | 13D: Mixed | 4 | 12: perf(4) + inp(3) + rt(2) + mem(3) | fraction | ✓ |

Mixed = numeric and categorical hyperparameters; perf = performance measures; rt = train/predict time; mem = memory consumption; inp = interpretability measures; H = Hierarchical search space. We do not include the fidelity parameter in the search space dimensionality.

A scenario is therefore a collection of instances sharing the same search space and objective(s), e.g., allowing for hyperparameter transfer learning between instances of the scenario. *Benchmark Suites* in turn are sets of instances that do not need to share the same objectives, but instead can consist of instances stemming from different scenarios.

## 4 YAHPO Gym

Motivated by the need for efficient and faithful benchmarks for HPO, we develop YAHPO Gym based on a set of *Criteria for HPO Benchmarks* discussed in Supplement B.1. YAHPO Gym is explicitly designed to use surrogate-based benchmarks only. It consists of a collection of 14 *scenarios* that can be evaluated across a total of $\sim 700$ instances. Each benchmark instance consists of an objective function that is parameterized in the form of a ConfigSpace Python object [48], making the search space computer-readable and readily usable with a range of existing HPO implementations. The objective function generates a prediction using the instance surrogate model, which is a compressed neural network. Table 2 provides an overview of all benchmark scenarios available in YAHPO Gym. We describe data sources as well as the full search spaces in Supplement F. We want to highlight the *rbv2_super* collection, which reflects an AutoML pipeline: It is, to our knowledge, the first available benchmark simulating a combined algorithm and hyperparameter selection problem [68] in the form of a high dimensional hierarchical search space by introducing the algorithm as an additional tunable hyperparameter.

In YAHPO Gym, every scenario allows for querying objective values at lower fidelities, enabling efficient benchmarking of multi-fidelity HPO methods. Analogously, every benchmark allows for returning multiple target metrics as criteria, enabling benchmarking of multi-objective HPO methods. Finally, almost all benchmark scenarios provide problems on a large number of instances (mostly ranging from 34 to 119), allowing for benchmarking of transfer-learning HPO methods. Predictions as well as sampling can be made reproducible through seeding. In order to achieve *portability* while still being *efficient*, YAHPO Gym uses fitted neural networks compressed via ONNX [3] as surrogate models. Our neural networks are ResNets for tabular data [26] consisting of up to 8 layers with a width of up to 512 and hyperparameters individually tuned for each scenario. We refer the reader to Supplement D for details regarding architecture and fitting procedure. Surrogate models have very small memory and inference time overhead and are compatible

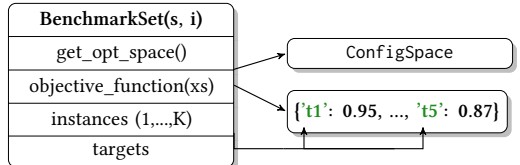

(a) YAHPO Gym's core functionality (**s**: scenario, **i**: instance, **xs**: configuration). Evaluating `objective_function` for a given configuration **xs** returns a dictionary of predicted metrics for a given scenario and instance.

```python
from yahpo_gym import *
b = BenchmarkSet('lcbench', instance='3945')
# Sample a point from the ConfigSpace
xs = b.get_opt_space().sample_configuration(1)
# Evaluate the configuration
b.objective_function(xs)
```

(b) Python code for instantiating a benchmark instance, sampling a new configuration and evaluating the objective function.

Figure 1: API overview.

across platforms and operating systems. In contrast to other benchmarks, evaluating $c(\lambda)$ requires only $10 - 100$ ms and only 100 MB of memory. In fact, YAHPO Gym's current infrastructure is so lightweight, it can easily be integrated in any existing toolbox or benchmark suite.

## 4.1 Suites: YAHPO-SO & YAHPO-MO

Together with YAHPO Gym, we propose two carefully selected *benchmark suites*. They constitute a proposal for surrogate-based benchmarks of HPO problems. We call those YAHPO-SO (single-objective, 20 instances) and YAHPO-MO (multi-objective, 25 instances). Together with the set of instances, we provide specific evaluation criteria, such as the budget available for optimization and number of stochastic replications as well as metrics to be used and fully specified search spaces which can be obtained from our software. Instances were selected across all scenarios taking into account approximation quality of the underlying surrogate and diversity. We consider those benchmarks a first draft for such a benchmark set (version **v1.0**) and explicitly invite the community to jointly work on a larger, more comprehensively evaluated set of benchmark instances. Details with respect to how instances were selected, and a full list of included instances, can be found in Supplement C.2. We conduct a benchmark providing anytime performance for a large variety of baselines on the proposed benchmark suites.

## 5 Tabular or Surrogate Benchmarks?

Consider the true objective $c(\lambda)$ of a *real* benchmark instance with $c : \tilde{\Lambda} \to \mathbb{R}$ in the single-objective setting. In a *tabular* benchmark, the domain of the objective function is implicitly discretized into a finite grid $\tilde{\Lambda}_{\text{discrete}}$ of the original domain and pre-evaluated at these points and the benchmark objective $\hat{c}_{\text{tabular}}(\lambda)$ is thus the original $c(\lambda)$ restricted to $\tilde{\Lambda}_{\text{discrete}}$. The extent to which discretization affects the faithfulness of tabular benchmarks depends on the nature and dimensionality of the search space: It disregards local structure in the response function and might even impose fixed fidelity schedules, should evaluations not be available at all budget levels. In order to assess the magnitude of this effect, we investigate the practical effects of discretization in the following experiment by comparing 8 black-box optimizers on *tabular*, *surrogate* and *real* versions of 5 synthethic multi-fidelity functions of varying dimensionality (Branin2D, Currin2D, Hartmann3D/6D, and Borehole8D [35]). The tabular benchmark is constructed by drawing and evaluating $10^6$ points from a grid. Surrogates are then fitted using those points. We compare Random Search (RS), several versions of Bayesian optimization (BO) and Hyperband (HB, [44]) across all settings. BO is configured with algorithm surrogate model either a Gaussian process (BO_GP), ensemble of feed-forward neural networks (BO_NN, [73]) or random forest (BO_RF, [12]) and acquisition function optimizer either Nelder-Mead/exhaustive search[3] (*_DF [54]) or Random

---

[3]for tabular benchmarks

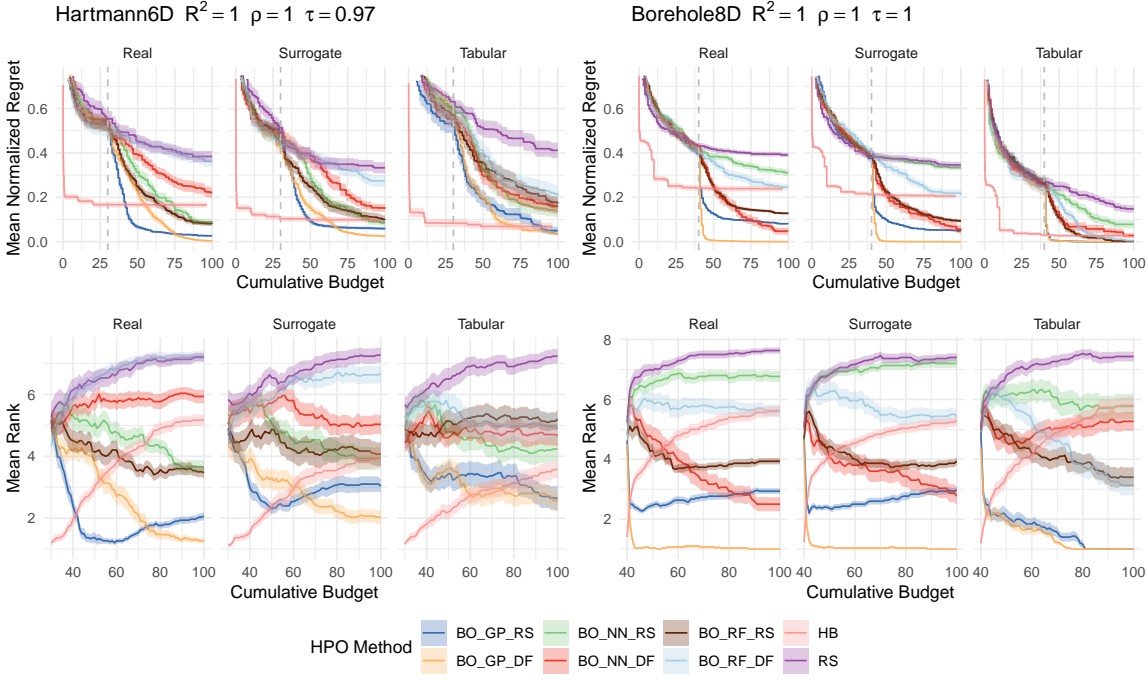

Figure 2: Mean normalized regret (top) and mean ranks (bottom) of different HPO methods on different benchmarks. Ribbons represent standard errors. The gray vertical line indicates the cumulative budget used for the initial design of BO methods. Performance measures of the surrogate benchmarks are stated after the benchmark function. 30 replications.

Search (*_RS). We describe additional details regarding the benchmark setup in Supplement E.1 and briefly present results: Figure 2 shows the anytime performance and mean rank of each HPO method split for the real, surrogate, and tabular benchmark on the Hartmann6D and Borehole8D test functions. We observe very similar performance traces of HPO methods on surrogate versions of benchmarks compared to real versions (Figure 2, top). However, in tabular benchmarks, we notice that for some problems, the BO methods converge substantially faster to a lower mean normalized regret (especially for BO_GP_*), which can possibly be explained by the much simpler infill optimization problem solved in the tabular case. Moreover, Hyperband appears to consistently perform better on tabular benchmarks. We further investigate average rankings over all replications (Figure 2, bottom). Each benchmark function yields an average ranking of HPO methods (e.g., with respect to final performance). Using consensus rankings, we can arrive at a single ranking over all benchmark functions [51] for a given benchmark type. We use the optimization based symmetric difference (SD) [36] minimizing rank reversals to compare both the surrogate and tabular inferred consensus rankings with the "ground truth" real function consensus ranking. We observe that consensus rankings obtained using surrogate benchmarks (permutation order 2) match more closely than tabular benchmarks (permutation order 5). We again provide additional details in Supplement E.1.

## 6  A Benchmark of HPO Methods on YAHPO Gym

We now demonstrate how YAHPO Gym can be used in practice to benchmark different HPO methods. We benchmark 7 single-objective HPO methods on YAHPO-SO and 7 multi-objective HPO methods on YAHPO-MO and want to answer the following research questions: (**RQ1**) *Do multi-fidelity (single-objective) HPO methods improve over full-fidelity methods?* (**RQ2**) *Do advanced multi-objective HPO methods improve over Random Search?*

## 6.1 RQ1: Do multi-fidelity (single-objective) HPO methods improve over full-fidelity methods?

We compare Random Search and SMAC (SMAC4HPO facade; [47]) to the multi-fidelity methods Hyperband [44], BOHB [21], DEHB [4], SMAC-HB (SMAC4MF facade; [47]) and optuna ([2]; TPE sampler and median pruner following successive halving steps). More details on the experimental setup and HPO methods is given in Supplement E.2. All optimizers are run for a total budget of $\lceil 20 + 40 \cdot \sqrt{\text{SEARCH\_SPACE\_DIM}} \rceil$ full-fidelity evaluations with 30 replications. Figure 3a shows the average rank of HPO methods with respect to their anytime performance. Figure 3b and Figure 3c show critical difference plots ($\alpha = 0.05$) of mean ranks after 25% and 100% of the optimization budget. The corresponding Friedman tests indicate significant differences ($p < 0.001$) in both cases. We observe that all multi-fidelity optimizers outperform Random Search with respect to intermediate performance (25% of optimization budget) and optuna, BOHB, SMAC-HB and Hyperband also outperform SMAC. With respect to final performance, SMAC takes the lead closely followed by SMAC-HB with other multi-fidelity optimizers slightly falling behind. We conclude that multi-fidelity HPO methods indeed improve over full-fidelity methods, but only with respect to intermediate performance. Our results are in line with what has been reported in other benchmarks [19] with the exception that optuna seems more competitive in our benchmark, while DEHB is less competitive. One reason for this difference might be that we include hierarchical search spaces in contrast to previous work.

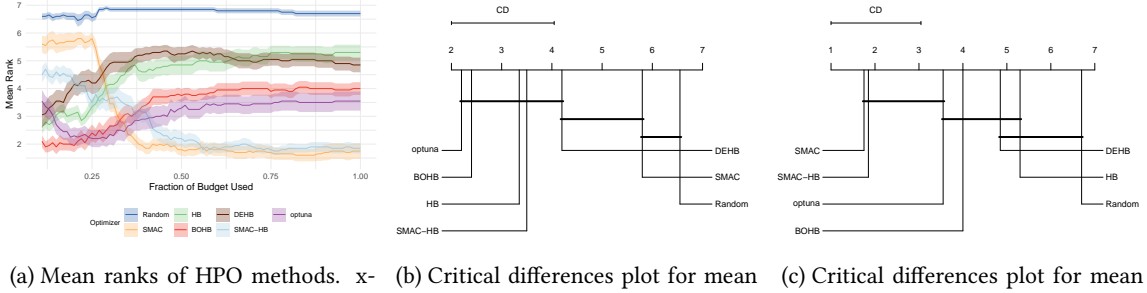

(a) Mean ranks of HPO methods. x-axis starts after 10%.

(b) Critical differences plot for mean ranks of HPO methods after 25% of the optimization budget.

(c) Critical differences plot for mean ranks of of HPO methods after 100% of the optimization budget.

Figure 3: Results of YAHPO-SO single-objective benchmark across 7 optimizers (20 instances).

## 6.2 RQ2: Do advanced multi-objective HPO methods improve over Random Search?

We compare Random Search, Random Search x4 (Random Search with quadrupled budget as a strong baseline), ParEGO [40], SMS-EGO [61], EHVI [20], MEGO [33] and MIES [46] on multi-objective HPO problems with $2 - 4$ objectives. More details on the experimental setup and HPO methods is given in Supplement E.3. All optimizers are run for a total budget of $\lceil 20 + 40 \cdot \sqrt{\text{SEARCH\_SPACE\_DIM}} \rceil$ full-fidelity evaluations for 30 replications. Figure 4a shows the average rank of HPO methods with respect to their anytime performance (determined based on the normalized Hypervolume Indicator). Figure 4b and Figure 4c show critical difference plots ($\alpha = 0.05$) of these ranks after 25% and 100% of the optimization budget. The corresponding Friedman tests indicate significant differences ($p < 0.001$) in both cases. We observe that not all methods significantly improve over Random Search with respect to final performance, i.e., EHVI and SMS-EGO fail to do so. Especially with respect to intermediate performance (25% of optimization budget), Random x4 outperforms all competitors. However, with respect to final performance, MEGO, ParEGO and MIES yield similar performance catching up to Random x4. We conclude that, in general, advanced multi-objective HPO methods improve over Random Search but also want to highlight that optimizer performance strongly varies with respect to the different benchmark instances.

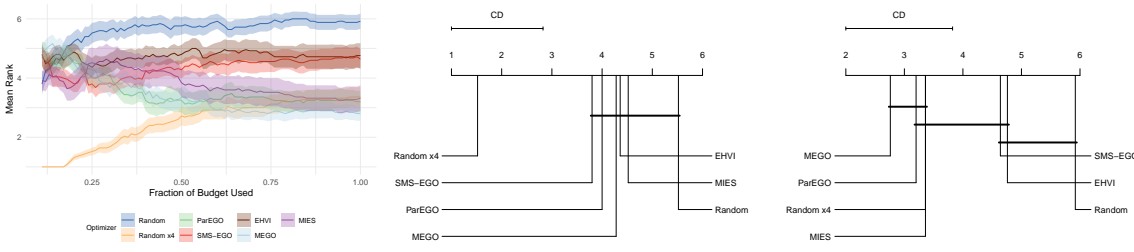

(a) Mean ranks of HPO methods. x-axis starts after 10%.

(b) Critical differences plot for differences in ranks of HPO methods after 25% of optimization budget.

(c) Critical differences plot for differences in ranks of HPO methods after 100% of optimization budget.

Figure 4: Results of the YAHPO-MO multi-objective benchmark across 7 optimizers (25 instances).

In total, both benchmarks described in this section took the equivalent of 139.57 CPU days using YAHPO Gym. We estimate that the YAHPO-SO benchmark, would take 14.75 CPU **years** when running real benchmarks, while our benchmark using YAHPO Gym took only 397.51 CPU **hours**, essentially speeding up evaluation by a factor of $\sim 300$.

## 7 Conclusions, Limitations and Broader Impact

We present YAHPO Gym, a multi-fidelity, multi-objective benchmark for HPO. Our benchmark is based on surrogates, which strike a favorable trade-off between faithfulness and efficiency, which we demonstrate in various experiments throughout our paper before conducting a large scale benchmark of modern single- and multi-objective optimizers. An as of yet under-explored domain are asynchronous optimization algorithms, which have recently gained popularity [45]. This has been studied in surrogate-based benchmarks by predicting runtimes and pausing the objective function for the predicted runtime, lowering computational demand for benchmarks but leading to a large waiting time [21]. In future work we plan on introducing faster-than-real time asynchronous benchmarking based on predicted runtimes.

**Limitations**. YAHPO Gym is based on surrogate models and therefore heavily relies on the faithfulness of those models in order to allow for valid conclusions. We have comprehensively evaluated surrogate models and provide a detailed report of performance metrics, hoping to demonstrate the faithfulness of our surrogates, but can only do so to a certain degree. We are furthermore aware that the real HPO problems modeled in our surrogates are in fact stochastic, and results can vary depending on randomness of the fitting procedure, data splits or initialization. We therefore provide a set of *noisy* surrogate models that intend to model the stochasticity of the problems using an ensemble of neural networks, but simultaneously allow for full control of the stochastic process by using random seeds.

**Broader Impact**. This manuscript presents a set of surrogate-based benchmarks for HPO. As such, our work does not have direct implications on society or individuals, but can lead to such indirectly if new methods are developed based on it. We would like to emphasize the possible **societal & environmental benefits**. First, we hope our benchmarks can improve the state of benchmarking in hyperparameter optimization contexts, leading to better tracking of progress in the discipline. Second, and more important, we hope that experiments based on YAHPO Gym can drastically reduce **computational cost** of hyperparameter optimization experiments. This type of experiments is usually extremely expensive, if real experiments are run for the evaluation of each HPC, which can be sped up by large factors if cheap approximations through surrogates are available.

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

## A  Maintenance of YAHPO Gym

Following [19], we present a maintenance plan for YAHPO Gym.

- Who is maintaining the benchmarking library?
  YAHPO Gym is developed and maintained by the *Statistical Learning and Data Science Group at LMU Munich.*

- How can the maintainer of the dataset be contacted (e.g., email address)?
  Questions should be submitted via an issue on the Github repository at `https://github.com/slds-lmu/yahpo_gym`.

- Is there an erratum?
  No.

- Will the library be updated?
  We plan on adding new instances as well as continuously updating existing instances should need occur. Changes will be communicated via Github releases as well as a *CHANGELOG*.

- Will older versions of the benchmarking library continue to be supported/hosted/maintained?
  Old versions are available via GitHub releases in the git repositories. We aim to support old versions on a best-effort basis with limited support for older versions.

- If others want to extend/augment/build on/contribute to the dataset, is there a mechanism for them to do so?
  We have detailed how additional benchmarks can be added in the documentation `https://slds-lmu.github.io/yahpo_gym/extending.html`. We have furthermore made available the full code used to tune, fit and export surrogate models used in YAHPO Gym. The code is easily extendable for future datasets.

- Which dependencies does YAHPO Gym have?
  YAHPO Gym currently relies on the following dependencies
  (versions used throughout experiments in brackets):

  - onnxruntime (1.10.0)

  - pyyaml (5.4.1)

  - configspace (0.4.20)

  - pandas (1.3.5)

## B  Benchmark Suites

### B.1  Criteria for Benchmark Suites and Instances

To allow for a more systematic assessment of the quality of benchmarking instances, we define criteria that guided the development of YAHPO Gym and which should be satisfied to make a compelling argument for the use of any HPO benchmark.

    I. **Representativity & Diversity of Tasks** The goal of benchmark suites is to allow for a ranking of HPO methods according to their performance on future problems. Instances should therefore cover response surfaces encountered in relevant problem domains.

    II. **Difficulty and Structure** Benchmarks must be non-trivial, i.e., they should contain instances of sufficient difficulty to identify rankings between optimizers. Search spaces should reflect

search spaces that are encountered frequently in practice including mixed spaces with inter-actions as well as hierarchical spaces and sufficient dimensionality.

III. **Faithfulness** Rankings based on approximations (e.g., for *tabular* and *surrogate* instances) should reflect true rankings. The performance of surrogate models $\hat{g}$ should be close enough to $g$ based on performance metrics such as Spearman's $\rho$.

IV. **Efficiency** Benchmark experiments often require repeated evaluation of several optimizers across several datasets leading to considerable computational (and consequentially environ-mental cost [63]). Benchmarks should therefore strive for computational efficiency.

V. **Ease of use** Benchmark software needs to be accessible and portable across operating systems and programming languages. In practice, systems which do not require complex set up or es-tablishment of databases might lead to more widespread adoption. Meta-data such as search spaces should be available and machine-readable. As benchmarks allow for embarrassingly parallel execution, parallelization should be supported.

VI. **Reproducibility** While performance estimation in practice often includes stochastic compo-nents, it is important that benchmark suites can be made reproducible through the use of random seeds. Additionally, software dependencies and versions should be clearly commu-nicated and design components should be fixed and versioned to avoid cherry picking.

VII. **Stochasticity** Performance estimates obtained in real instances are realizations of random variables. In order to reflect this in practice, instances should allow for repeated evaluations.

While we consider the above requirements for good benchmarking suites, we furthermore want to highlight other properties that might be relevant for benchmarking suites.

A. **Multi-fidelity** Multi-fidelity methods have been shown to considerably speed up evaluation. Benchmark instances should therefore allow for querying performances at multiple fidelities.

B. **Runtime** In practice, HPO evaluations, especially for complex AutoML scenarios, can have very heterogeneous runtimes [66], which should also be reflected in a realistic benchmark by providing access to (estimated) runtimes which could subsequently be used to more accurate benchmark cost-efficient optimization methods.

C. **Asynchronous Evaluation** Although technically non-trivial, benchmarks should ideally allow the comparison of parallel HPO methods, allowing to compare, e.g., asynchronous HPO pro-cedures [45, 39].

D. **Multi-Objective** In many scenarios, users are not only interested in maximizing a single perfor-mance metric such as accuracy, but instead multiple relevant metrics such as calibration, infer-ence time, memory usage, and many others. We therefore consider including multi-objective HPO problems an important characteristic of a benchmark suite.

E. **Meta-Learning** Last but not least, in many cases, data collections are used to test scenarios for *meta-learning* [70, 59, 24] or *transfer learning* [75, 58]. For these scenarios, the availability of data across a large amount of datasets is often useful.

### B.2 Comparison to other Benchmark Suites

While a variety of benchmarking suites for optimization such as COCO [29], HPOlib [16], ASlib [11] and others exist, we do not go into detail and instead refer the reader to [19] where those libraries are discussed in more detail. We instead compare YAHPO Gym to the most similar suites: HPOBench [19] and HPO-B [60] and discuss and justify assessments made in Table 1.

Evaluations in Table 1 follow the doctrine *"the documentation is the product"* and we therefore consider only features that are explicitly documented in the accompanying manuscript and doc-

umentation, not considering other features. We note that all three libraries could theoretically be used or extended for additional tasks such as multi-objective evaluations but instead focus on scenarios where the considered property is explicitly included in the documented API. We furthermore note that several important aspects such as ease of use are not easily quantifiable and assessments made are therefore subjective. We derive assessments made in this section based on the criteria defined in Supplement B.1.

I. **Representativity** YAHPO Gym contains 14 across diverse search spaces for widely used ML algorithms trained on representative datasets. Search spaces are often mixed and sometimes include dependent hyperparameters resulting in a hierarchical search space. While theoretically possible, none of the instances in HPOBench currently contain hierarchical search spaces. HPO-B only supports continuous search spaces.

II. **Difficulty** To the best of our knowledge, it is not yet clear how to assess the difficulty of a benchmark instance. We therefore instead focus on showing that benchmark instances in YAHPO Gym are not trivial, e.g., constant across the full search space.

III. **Faithfulness** We evaluate the quality of fitted surrogates in Supplement D.2. To the best of our knowledge, analyses that establish the faithfulness of tabular benchmarks have not been conducted for tabular benchmarks previously.

IV. **Efficiency** We consider efficiency with respect to two aspects: *computational cost* and *memory consumption*. Tabular benchmarks often keep the full data in memory, essentially limiting the amount of parallel optimization runs on a given hardware required, e.g., for replications of stochastic benchmark experiments. Moreover, surrogate benchmarks are often based on un-optimized models fitted for each single instance. As a result, the required metadata (and memory consumption when multiple models are kept in memory) is often comparatively large. Our surrogates in contrast are highly optimized, compressed neural networks fitted across an entire scenario. Our surrogates are furthermore portable across platforms, alleviating concerns regarding software dependencies. Prediction on a surrogate requires only 10-100 ms and around 100 MB of memory allowing for a high degree of parallelization. In a small experiment, we estimate runtime and memory overhead for 300 iterations of Random Search on comparable SVM search spaces in Table 1 using the Python memory profiler (`https://pypi.org/project/memory-profiler/`). Since memory profiling is not accurate for HPO-Bench due to external processes, we estimate memory consumption using `htop`. Differences partially stem from more expensive setup in other libraries, but we consider 300 iterations of Random Search a representative use-case for many scenarios. Benchmarks were conducted on an AMD Ryzen 5 3600 6-Core CPU.

V. **Ease of use** YAHPO Gym does not require setting up containerization or any database and has only four dependencies that are both widely used and mature. All metadata required can be downloaded from a single, versioned metadata repository [4]. The modules API is simple to use (see, e.g., Figure 1). Other benchmarking suites either require *benchmark instance specific* software dependencies that can differ from benchmark instance to instance. While HPOBench has solved this using *containerization* adding considerable computational overhead, our surrogates only rely on a single fixed version of `ONNX` and can therefore completely ignore the problem.

VI. **Reproducibility** surrogates used in the benchmarking suites proposed along with YAHPO Gym are deterministic. Reproducibility therefore only requires ensuring seeding of any stochastic procedures in the optimization algorithm. Furthermore, we fix several design choices that might lead to differences between benchmarks: *i*) search spaces $\tilde{\Lambda}$ are fixed

---

[4]`https://github.com/slds-lmu/yahpo_data`

for each scenario and should be used in benchmarks *ii*) target metrics and exact evaluation protocol are fixed within the benchmark suites (see Supplement C.2) to ensure comparability.

Additional properties $A. - E.$ described in Supplement B.1 are compared in Table 1 and described in more detail below.

A. **Multi-fidelity** Only surrogate based benchmarks allow doing so for the full range of available fidelity steps. This essentially enforces evaluation at fixed fidelities in tabular benchmarks, e.g., disallowing evaluation of differing fidelity schedules. In contrast, surrogates in YAHPO Gym allow for evaluation at all fidelity steps.

B. **Runtime** All surrogates in YAHPO Gym allow for querying the predicted runtime for training a configuration, essentially allowing benchmarking methods that take into account runtimes.

C. **Asynchronous Evaluation** To our knowledge, none of the existing benchmark suites allow for asynchronous evaluation (except for *real* instances in *HPO-Bench*). YAHPO Gym currently allows for asynchronous evaluation, but this is considered an experimental feature. We hope to be able to fully allow asynchronous benchmarking in future versions of our benchmark.

D. **Multi-Objective** YAHPO Gym explicitly includes multiple objective for each scenario and allows the user to subset the returned targets explicitly. In contrast, HPO-Bench contains only few multi-objective benchmarks and does not explicitly document how they are supposed to be used.

E. **Transfer Learning** All considered suites allow for transfer learning. In contrast to *HPO-Bench* and *HPO-B*, YAHPO Gym includes the (to our knowledge) largest collection of instances for a given scenario for the *rbv2_\** scenarios consisting of up to 119 instances. Only few collections in HPOBench contain enough instances for meta-learning.

We furthermore define a *single objective* as well as a *multi-objective* benchmark task that include a evaluation protocol with respect to instances, search spaces, evaluation budget and target metrics. This allows for reproduction and extension by practitioners without additional design choices and provides a singular point of references.

## B.3 A Benchmark Instance

In order to improve differentiation, we formally define four different types of benchmark instances derived from Definition 1. We therefore only consider benchmarks based on *tabular, surrogate* and *real* instances in our manuscript.

**Definition 3 (Synthetic Benchmark Instance)** *A synthetic benchmark instance is a benchmark instance, where $g : \boldsymbol{\lambda} \to \mathbb{R}^m$ is a mathematically tractable function.*

*Synthetic* instances, such as the ones, e.g., included in *COCO* [29] rely on mathematically tractable test functions (e.g., Rosenbrock-2D) as response surface. While they provide cheap evaluations, problem structures in such functions are qualitatively distinct from test functions encountered in HPO scenarios, and the resulting optimization problem is therefore often not representative for optimization problems typically encountered in HPO.

**Definition 4 (Tabular Benchmark Instance)** *A tabular benchmark instance returns function evaluations $g(\boldsymbol{\lambda})$ from a table of pre-recorded performance results. Performance results are typically obtained by estimating $\widehat{\mathrm{GE}}(\mathcal{I}, \mathcal{J}, \rho, \boldsymbol{\lambda})$ for given $\mathcal{I}, \mathcal{J}$ and $\rho$. In contrast to synthetic and surrogate instances, the search space $\Lambda$ is discretized and $g$ can therefore be only evaluated at discrete points $\tilde{\Lambda} \in \Lambda$.*

**Definition 5 (Surrogate Benchmark Instance)** *A surrogate benchmark returns predictions $\hat{g}(\lambda)$ of machine learning models trained to infer the functional relationship between $\lambda$ and function evaluations $g(\lambda)$ based on a set of pre-recorded performance results.*

For clarity, we would like to differentiate in terminology between the *instance surrogate* of a surrogate benchmark, and the algorithm surrogate potentially used by an HPO method, e.g., the Gaussian process as surrogate model in BO explicitly mentioning the algorithm surrogate where required. The instance surrogate model $\hat{g}$ or the tabular data should approximate the true relationship between $\lambda$ and the target metrics reasonably well. We consider a mapping $\hat{g}$ to be *faithful* if:

1. cross-validated performance metrics are sufficiently good with respect to metrics such as $R^2$ and Spearman's $\rho$. We typically consider a cutoff $\rho > 0.7$ for including a surrogate.

2. if the induced ranking of optimizers on a given $\hat{g}$ closely resembles the true rankings on the original underlying optimization problem (in general, the *real* setting relying on $g$).

3. learning curves of HPO methods on $\hat{g}$ closely resemble the true performance curves.

**Definition 6 (Real Benchmark Instance)** *A real benchmark instance returns function evaluations $g(\lambda)$. Performance results are typically obtained by estimating $\widehat{\mathrm{GE}}(\mathcal{I}, \mathcal{J}, \rho, \lambda)$ for given $\mathcal{I}, \mathcal{J}$ and $\rho$.*

Since the same benchmark instance can be provided as a *real*, *tabular*, or *surrogate* instance, we speak of different *versions* of that instance where required.

## C YAHPO Gym

In the following we will provide additional details on general aspects of YAHPO Gym. A detailed description of included surrogates can be found in Supplement D and a detailed description of used data and included search spaces can be found in Supplement F.

### C.1 Usage

The `yahpo_gym` software can be directly installed from *GitHub*[5] and only requires downloading one additional GitHub repository containing metadata[6] in an initial setup step.

#### HPO Benchmarking

To ensure interoperability with different optimizer API's, YAHPO Gym offers only evaluation of the objective function using the `BenchmarkSet.objective_function(xs)` method (where `xs` is a hyperparameter configuration to be evaluated). This allows for use with many different optimizers (see, e.g., examples provided in the accompanying notebooks). We furthermore allow for querying the search space using `BenchmarkSet.get_opt_space(xs)` in order to ensure that optimizers are ran on comparable search spaces. We provide additional details with respect to exact setups.

#### Transfer HPO

Different forms of Transfer HPO are available in YAHPO Gym and can be setup analogous by querying the objective function across different instances of the scenario. We present examples in the modules documentation.

---

[5] https://github.com/slds-lmu/yahpo_gym
[6] https://github.com/slds-lmu/yahpo_data

### C.2 Benchmark Suites: YAHPO-SO & YAHPO-MO

This section provides additional details with respect to the two benchmark sets proposed with YAHPO Gym. Both suites can be obtained via `get_suites(<type>, <version>)` specifying the type of the benchmark (currently supporting "single" for YAHPO-SO and "multi" for YAHPO-MO) and the version (currently 1.0).

- Optimizers should use the search spaces included in YAHPO Gym in order to establish that differences in performance do not depend on differing search spaces.

- Optimization should be run for $\lceil 20 + 40 \cdot \sqrt{\text{SEARCH\_SPACE\_DIM}} \rceil$ steps. Each step is equivalent to a full budget evaluation, essentially allowing multi-fidelity method the same number of full budget equivalents. We report the budgets for each scenario in Table 3 and Table 4.

- Target metrics to be used with the single-objective and multi-objective suite are reported in Table 3 and Table 4.

- We encourage reporting *mean normalized regret* and *mean ranks* for the anytime performance of an optimizer. Reported values are based on the target metric for YAHPO-SO and the normalized Hypervolume Indicator for YAHPO-MO.

- In order to assess variance, we encourage reporting averages and standard errors across 30 replications with differing random seeds.

We will now go on to discuss criteria for inclusion of tasks in the respective benchmarks.

In light of the criteria defined in Supplement B.1, we strive for diversity by including instances from all included scenarios. We consider only surrogates that are *faithful* (measured via Spearman's $\rho$ reported for each target below). Our benchmarks are made available through a fully documented API. Inference on a surrogate model is highly efficient taking usually only 10-100 milliseconds per batch. Benchmarks are furthermore reproducible and allow for parallelization and runtime prediction on a continuous range of fidelities. We include search spaces for all problems in Supplement F.

We furthermore briefly want to discuss selecting a budget that depends on the scenario at hand. We consider the search space dimension to be a relevant input for determining the overall optimization budget that should be used for optimization. Our formula ensures, that optimization runs for a minimum of 77 iterations (iaml_glmnet, 2D) and a maximum of 267 (rbv2_super, 38D) iterations, which we consider useful bounds for the respective search space dimensionality, especially given that multi-fidelity allows for evaluations at a fraction of the full budget.

### C.3 R package

While we focus on the Python module in the manuscript, YAHPO Gym offers an R interface that is equivalent in functionality. We do not present the API in detail here since it follows the same principles and naming conventions as the Python module. Further information is available from the package documentation. Listing 1 contains the sample R-code used to first draw a random configuration from the search space and then evaluate the drawn configuration.

## D YAHPO Gym Surrogates

On an implementation level, YAHPO Gym consists of a (versioned) Python module / R package `yahpo_gym` and a (versioned) set of required metadata (including fitted surrogate models) which we will call `yahpo_data` in the following. The core contribution in YAHPO Gym is a set of surrogate

Table 3: **YAHPO-SO** (v1): Collection of single-objective benchmark instances. We indicate surrogate approximation quality using Spearman's $\rho$.

|  | Scenario | Instance | Target | $\rho$ | Budget |
|---|---|---|---|---|---|
| 1 | lcbench | 167168 | val_accuracy | 0.94 | 126 |
| 2 | lcbench | 189873 | val_accuracy | 0.97 | 126 |
| 3 | lcbench | 189906 | val_accuracy | 0.97 | 126 |
| 4 | nb301 | CIFAR10 | val_accuracy | 0.98 | 250 |
| 5 | rbv2_glmnet | 375 | acc | 0.80 | 90 |
| 6 | rbv2_glmnet | 458 | acc | 0.85 | 90 |
| 7 | rbv2_ranger | 16 | acc | 0.93 | 134 |
| 8 | rbv2_ranger | 42 | acc | 0.98 | 134 |
| 9 | rbv2_rpart | 14 | acc | 0.92 | 110 |
| 10 | rbv2_rpart | 40499 | acc | 0.97 | 110 |
| 11 | rbv2_super | 1053 | acc | 0.31 | 267 |
| 12 | rbv2_super | 1457 | acc | 0.70 | 267 |
| 13 | rbv2_super | 1063 | acc | 0.57 | 267 |
| 14 | rbv2_super | 1479 | acc | 0.36 | 267 |
| 15 | rbv2_super | 15 | acc | 0.75 | 267 |
| 16 | rbv2_super | 1468 | acc | 0.77 | 267 |
| 17 | rbv2_xgboost | 12 | acc | 0.93 | 170 |
| 18 | rbv2_xgboost | 1501 | acc | 0.89 | 170 |
| 19 | rbv2_xgboost | 16 | acc | 0.91 | 170 |
| 20 | rbv2_xgboost | 40499 | acc | 0.96 | 170 |

```r
library("yahpogym")
library("paradox")
library("bbotk")
# Instantiate the BenchmarkSet
b = BenchmarkSet$new('lcbench', instance='3945')
# Get the objective
objective = b$get_objective('3945', check_values = FALSE)
# Sample a point from the ConfigSpace
xdt = generate_design_random(b$get_search_space(), 1)$data
xss_trafoed = transform_xdt_to_xss(xdt, b$get_search_space())
# Evaluate the configuration
objective$eval_many(xss_trafoed)
```

Listing 1: R-code to sample and evaluate a configuration using YAHPO Gym.

models[7] based on neural networks. This section provides additional details with respect to the fitting procedures of surrogate models as well as a rigorous evaluation of the final surrogates.

## D.1 Setup and Training

Previous work [17, 18, 65] suggests that tree based regression methods such as random forests [12] are very suited as instance surrogate models for (single-objective) benchmarks. However, in YAHPO Gym we want to predict multiple target metrics for each instance of a benchmark collection efficiently and compactly. As a result, we use neural network surrogates because they 1) can naturally handle multiple outputs and do not require a model for each target metric and 2)

---
[7]available at https://github.com/slds-lmu/yahpo_data

Table 4: **YAHPO-MO** (v1): Collection of multi-objective benchmark instances. We indicate surrogate approximation quality using Spearman's $\rho$ (averaged over targets).

|   | Scenario | Instance | Targets | $\rho$ | Budget |
|---|----------|----------|---------|--------|--------|
| 1 | iaml_glmnet | 1489 | mmce,nf | 0.86 | 77 |
| 2 | iaml_glmnet | 1067 | mmce,nf | 0.73 | 77 |
| 3 | iaml_ranger | 1489 | mmce,nf,ias | 0.93 | 134 |
| 4 | iaml_ranger | 1067 | mmce,nf,ias | 0.92 | 134 |
| 5 | iaml_super | 1489 | mmce,nf,ias | 0.82 | 232 |
| 6 | iaml_super | 1067 | mmce,nf,ias | 0.82 | 232 |
| 7 | iaml_xgboost | 40981 | mmce,nf,ias | 0.88 | 165 |
| 8 | iaml_xgboost | 1489 | mmce,nf,ias | 0.92 | 165 |
| 9 | iaml_xgboost | 40981 | mmce,nf,ias,rammodel | 0.89 | 165 |
| 10 | iaml_xgboost | 1489 | mmce,nf,ias,rammodel | 0.92 | 165 |
| 11 | lcbench | 167152 | val_accuracy,val_cross_entropy | 0.98 | 126 |
| 12 | lcbench | 167185 | val_accuracy,val_cross_entropy | 0.91 | 126 |
| 13 | lcbench | 189873 | val_accuracy,val_cross_entropy | 0.93 | 126 |
| 14 | rbv2_ranger | 6 | acc,memory | 0.90 | 134 |
| 15 | rbv2_ranger | 40979 | acc,memory | 0.73 | 134 |
| 16 | rbv2_ranger | 375 | acc,memory | 0.85 | 134 |
| 17 | rbv2_rpart | 41163 | acc,memory | 0.85 | 110 |
| 18 | rbv2_rpart | 1476 | acc,memory | 0.80 | 110 |
| 19 | rbv2_rpart | 40499 | acc,memory | 0.83 | 110 |
| 20 | rbv2_super | 1457 | acc,memory | 0.66 | 267 |
| 21 | rbv2_super | 6 | acc,memory | 0.68 | 267 |
| 22 | rbv2_super | 1053 | acc,memory | 0.45 | 267 |
| 23 | rbv2_xgboost | 28 | acc,memory | 0.80 | 170 |
| 24 | rbv2_xgboost | 182 | acc,memory | 0.79 | 170 |
| 25 | rbv2_xgboost | 12 | acc,memory | 0.76 | 170 |

should scale better than a random forest (fitted on each target metric) if the dimensionality of the data (especially in the number of features) increases.

Surrogate models used in YAHPO Gym are based on `ResNet` architectures for tabular data [26]. Instead of relying on a fixed architecture, we tune the neural network for each *Scenario* using `optuna` [2]. We used the Adam optimizer for a maximum of 100 epochs (early stopping with patience of 10) with L2 loss. Surrogates were trained jointly for each benchmark scenario (for all instances and target metrics). We use a stratified train/validation/test split of 0.6/0.2/0.2, using the validation data to determine the surrogate model architecture and report performances on the test set. The search space as well as the fully reproducible code for fitting can be obtained at YAHPO Gym. Tuning and fitting of a single *Scenario* takes 3 GPU days on average on an NVIDIA DGX-A100 instance, we therefore estimate a one time cost of 45 GPU days for establishing the full benchmark.

We adapt the architecture proposed in [26] in multiple ways:

**Feature- and Output-Scaling** Hyperparameters as well as resulting performance metrics (e.g learning rates of log-loss values) often vary across orders of magnitudes. We have practically observed that transforming target metrics to the unit cube prior to training and reverse-transforming afterwards massively improves quality of the resulting surrogates. Available scaling techniques include *Neg-Exp* and *Log* transformation before scaling to [0, 1]. We furthermore include clamping to ensure that predictions are in valid ranges. Non-numeric features were transformed via entity embeddings [28].

**Ensembles** In order to allow for an estimate of variance, we make *noisy* versions of our surrogates available together with the standard *deterministic* set of surrogates. Ensembles consist of replications of the architecture determined during tuning and fitted on different permutations of the data with differing initial weights. The prediction step is the weighted average over predictions from ensemble members with weights $\alpha_i$ sampled from a Dirichlet distribution.

We additionally consider scenarios that allow simulating **asynchronous evaluation** and therefore predict the time of the training procedure using our surrogates. YAHPO Gym currently supports asynchronous scheduling by estimating the runtime of training a model and then idling the system for the estimated time. This is implemented via `objective_function_timed` in `yahpo_gym` but currently considered in an experimental status.
In future work, we hope to propose and evaluate a surrogate-based benchmark explicitly allowing for benchmarking of asynchronous scheduling strategies based on surrogate predictions. To enable more realistic scheduling, we hope to furthermore include memory constraints using predicted peak memory consumption for a training run.

## D.2 Surrogate Quality

We provide an overview over surrogate quality measured on the test set using Spearman's $\rho$ averaged across all instances in Table 5. Metrics are routinely $\geq 0.9$ except for few instances / target metrics and even surpasses performances for surrogate models reported, e.g., in [65]. We furthermore depict real and predicted learning curves for four randomly drawn configurations in Figure 5. Note that in our work, learning curves are predicted only based on hyperparameters, and not based on initial, low-fidelity observations (as done in learning curve prediction tasks). Our surrogates therefore solve a much harder task. Surrogates in general predict the learning curves with a high degree of precision.

Table 5: Average surrogate performance (Spearman's $\rho$) across all instances per scenario/target. We abbreviate cross_entropy (ce) and balanced_accuracy(bac) for brevity.

| Scenario | $\rho$ |
|---|---|
| iaml_glmnet | mmce:0.97,f1:0.9,auc:0.92,logloss:0.97,rammodel:0.97,timetrain:0.95,mec:0.9,ias:0.91,nf:0.97 |
| iaml_ranger | mmce:0.99,f1:0.98,auc:1,logloss:0.95,rammodel:1,timetrain:0.91,mec:0.88,ias:0.98,nf:1 |
| iaml_rpart | mmce:0.99,f1:0.96,auc:0.99,logloss:0.96,rammodel:1,timetrain:0.96,mec:0.71,ias:0.96,nf:0.96 |
| iaml_super | mmce:0.93,f1:0.95,auc:0.89,logloss:0.93,rammodel:0.71,timetrain:0.61,mec:0.94,ias:0.65,nf:0.92 |
| iaml_xgboost | mmce:0.97,f1:0.98,auc:0.97,logloss:0.93,rammodel:0.86,timetrain:0.71,mec:0.95,ias:0.84,nf:0.99 |
| lcbench | time:0.94,val_accuracy:0.95,val_ce:0.97,val_bac:0.98,test_ce:0.99,test_bac:0.98 |
| nb301 | val_accuracy:0.98,runtime:0.94 |
| rbv2_aknn | acc:0.99,bac:0.99,auc:0.98,brier:1,f1:0.91,logloss:0.99,timetrain:0.64,memory:0.83 |
| rbv2_glmnet | acc:0.99,bac:0.95,auc:0.91,brier:1,f1:0.96,logloss:0.99,timetrain:0.79,memory:0.82 |
| rbv2_ranger | acc:0.99,bac:0.98,auc:0.95,brier:1,f1:0.92,logloss:1,timetrain:0.84,memory:0.66 |
| rbv2_rpart | acc:0.98,bac:0.96,auc:0.93,brier:0.99,f1:0.93,logloss:0.98,timetrain:0.72,memory:0.86 |
| rbv2_super | acc:0.82,bac:0.78,auc:0.73,brier:0.91,f1:0.91,logloss:0.89,timetrain:0.69,memory:0.71 |
| rbv2_svm | acc:0.99,bac:0.98,auc:0.94,brier:0.99,f1:0.91,logloss:0.99,timetrain:0.76,memory:0.84 |
| rbv2_xgboost | acc:0.98,bac:0.96,auc:0.94,brier:0.99,f1:0.92,logloss:0.98,timetrain:0.93,memory:0.78 |

Some of the targets available require further study and we therefore discourage their use in benchmarks. Those are *rampredict & ramtrain* (iaml_* scenarios) as well as *timepredict* (rbv2_* scenarios). Reasons for this assessment are partially poor surrogates, but we also assume that the underlying data is at fault: Prediction times are often very small and heavily influenced by system load, while correct estimation of required memory are relatively difficult to obtain in general.

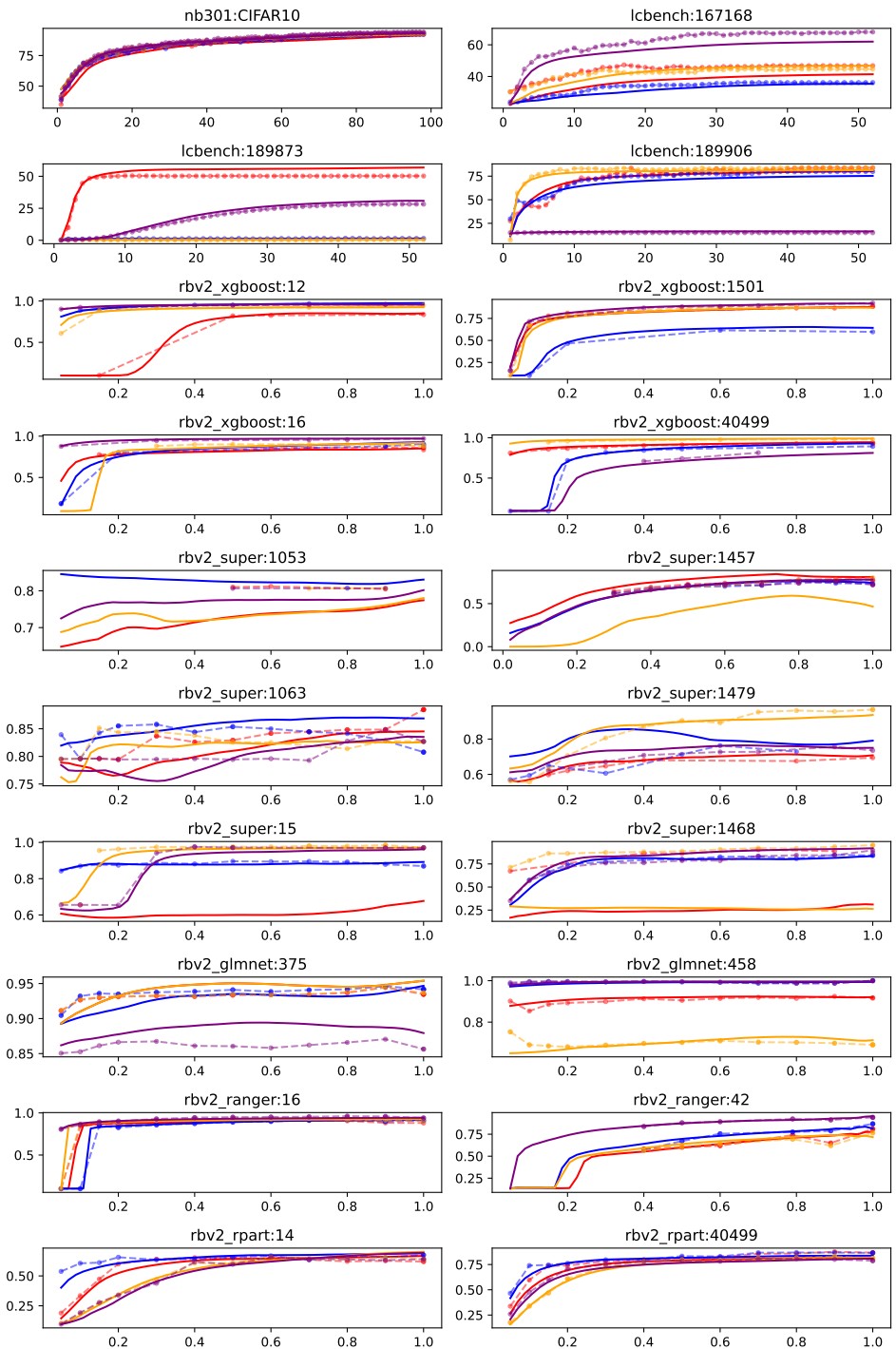

Figure 5: Predicted learning curves (lines) together with true learning curves (dotted) for four randomly drawn configurations (differentiated by colour) out of each instance in YAHPO-SO reporting the respective target metric.

## D.3 Instance Difficulty

We quantify difficulty of instances using the empirical cumulative distribution function (ECDF), assuming that difficult instances have only a small probability mass close to the optimum. ECDFs

for all instances in YAHPO-SO are shown in Figure 6. Differences between real evaluations and surrogate predictions can stem from the sampling procedure (random on surrogates vs. unknown sampling for real evaluations), as well as biases in the surrogates. All evaluations are made at maximal fidelity. We furthermore provide ECDF plots for all optimizers in Figure 7. This allows for a different perspective on the quality of solutions found by the different optimizers.

## E  Experiments

### E.1  Tabular vs. Surrogate Benchmarks

**Resolution of Tabular Benchmarks**. In practice, the resolution of grid points needs to be low for high dimensional spaces to limit the resulting table to a usable size. With purely categorical search spaces, often used in NAS, an exhaustive (i.e., $\tilde{\Lambda}_{\text{discrete}} = \tilde{\Lambda}$) tabular benchmark is often possible, as in, e.g., NAS-Bench-101 [76], which contains "only" 423k unique architectures. Multi-fidelity evaluations essentially add an additional dimension to the optimization problem when considering tabular data, since each evaluation now needs to be stored at multiple fidelity steps. If fidelity steps are not available at all budget levels, optimization benchmarks can be restricted to fixed fidelity progression (e.g., geometric progression as used in Hyperband).

**Discrete Search Spaces**. The modification of the search space from $\tilde{\Lambda}$ to $\tilde{\Lambda}_{\text{discrete}}$ can be handled in one of two ways: One can let HPO methods operate on the original search space $\tilde{\Lambda}$ and transparently "round" values to the nearest point contained in $\tilde{\Lambda}_{\text{discrete}}$. This effectively presents the optimization algorithm with a locally constant objective function. Alternatively, one can inform the HPO algorithm about the discrete nature of $\tilde{\Lambda}_{\text{discrete}}$, and possibly even modify the optimization procedure. As an example, consider the acquisition function optimization step within the BO framework: In the context of tabular benchmarks, the problem of optimizing the infill criterion becomes trivial because one can perform an exhaustive search over all points not yet evaluated to determine the next candidate(s) for evaluation. Note that we could also proceed to use a 1-Nearest-Neighbor model to evaluate HPCs in tabular benchmarks. This essentially results in a surrogate benchmark because we now rely on a performance model for the evaluation. In contrast to approximation by discretization, in a surrogate benchmark the domain of the objective function is not explicitly altered. Instead, predictions of an instance surrogate regression model $\hat{g}(\cdot)$ are returned as function evaluations, $\hat{c}_{\text{surrogate}} : \tilde{\Lambda} \to \mathbb{R}^m, \lambda \mapsto \hat{g}(\lambda)$. The drawback here is that values returned by the surrogate model may misrepresent the local structure of the problem as well. Beyond the resolution of the surrogate model training data, these structures are interpolated and influenced by the inductive bias implied by the model.

**Experimental Setup**. As a *real* benchmark, we consider the original synthethic benchmark function, while we generate a grid containing at most $10^6$ points for the tabular version, storing these pre-evaluated points in a look-up table together with their function value. The resolution of the grid is the same for all functions along the budget parameter dimension, with 10 grid points ranging from $2^{-9}$ to 1 on a $2^x$ scale. For all other parameters of the domain, an equidistant grid was generated by using $\lfloor (10^5)^{\frac{1}{D}} \rfloor$ grid points for each dimension $d = 1, \ldots, D$. With the same data we employ a similar surrogate neural network as used in YAHPO Gym. We compare the following methods on real, surrogate, and tabular benchmarks: All HPO methods were run for a total budget of 100 evaluations reflecting 100 full fidelity evaluations. The synthetic test functions used in the experiments [35] include a multi-fidelity parameter allowing for the use of multi-fidelity methods such as Hyperband. Of the methods investigated, only HB makes use of the fidelity parameter, while all other methods perform full budget evaluations. As a surrogate, we train a Wide & Deep Network [13]. More details can be found in `https://github.com/slds-lmu/yahpo_exps`. BO variants used Expected Improvement [34] as acquisition function and an initial design of $5 \cdot D$ points sampled uniformly at random. The Gaussian process surrogate model used a Matérn 3/2

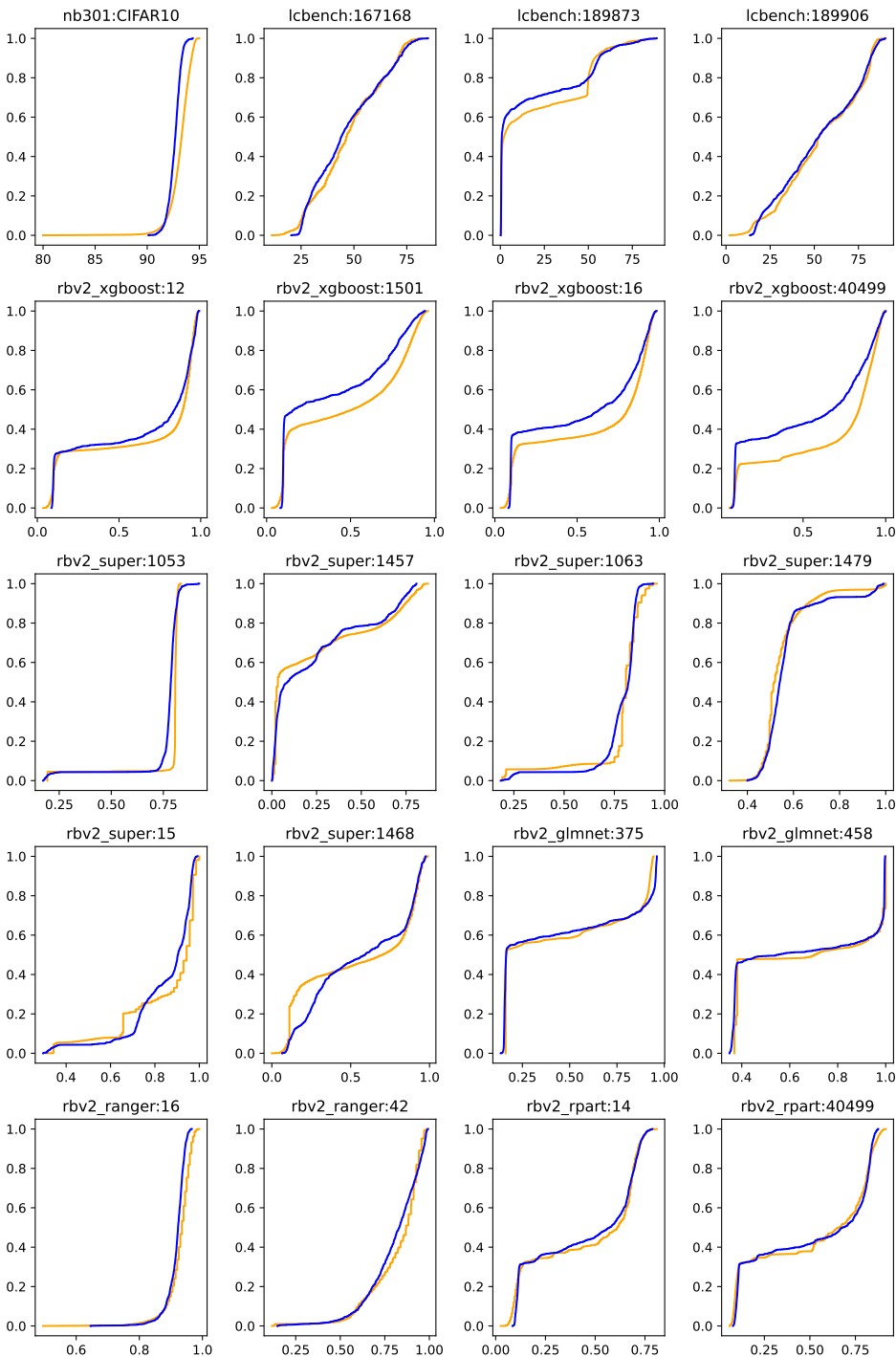

Figure 6: Empirical cumulative distribution function (ECDF) for surrogate predictions (blue) and real evaluations (orange).

kernel. Nelder-Mead as acquisition function optimizer was terminated if the relative change in the maximum fell below $1e - 4$. Tabular benchmarks used an exhaustive search for optimizing the acquisition function in the scenario of *_DF. Random Search as acquisition function optimizer

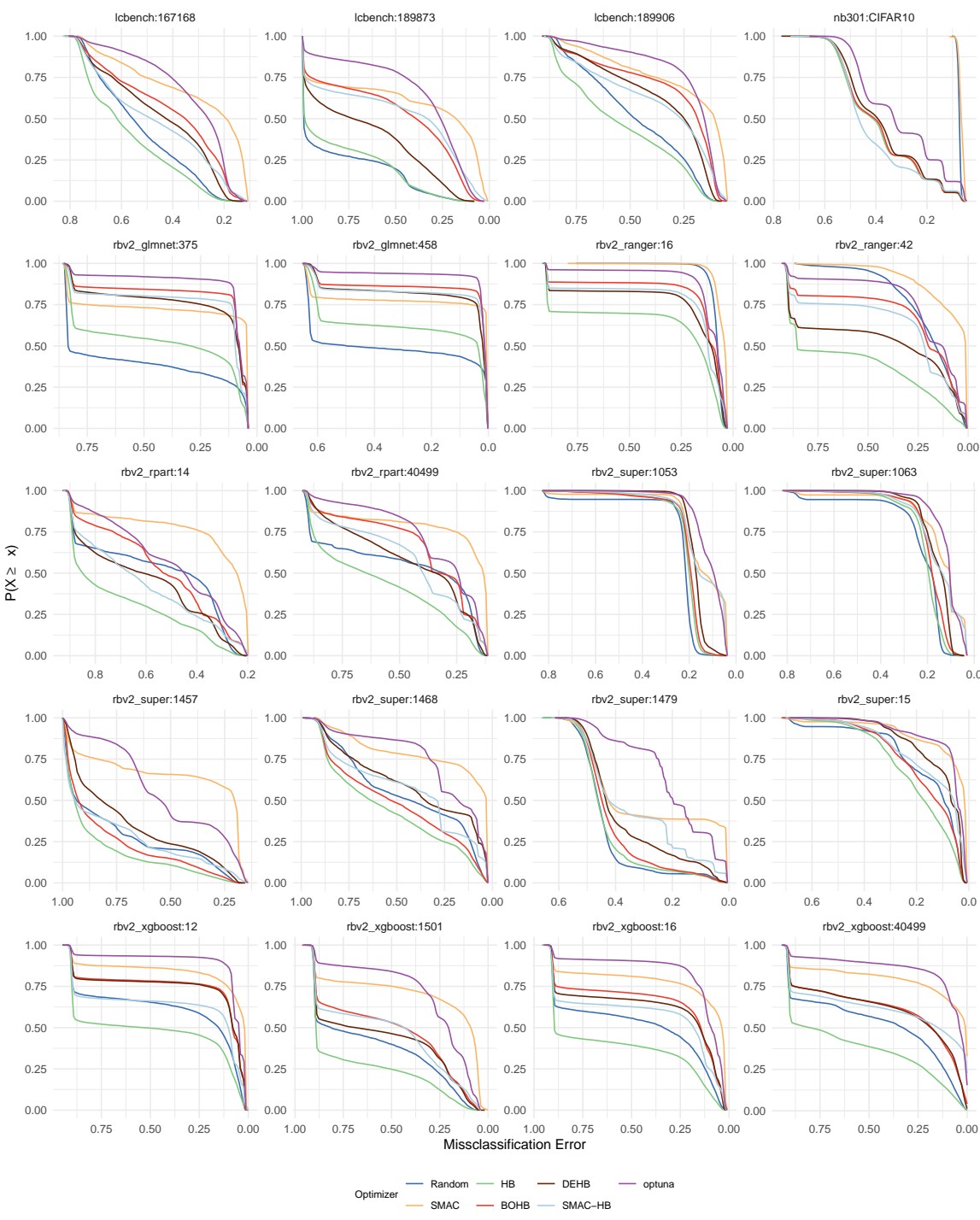

Figure 7: Empirical cumulative distribution function (ECDF) for optimizer traces on YAHPO-SO.

was allowed $10^4$ evaluations.

**Evaluation.** For evaluation, we computed the mean normalized regret for each HPO method separately on the real, surrogate and tabular benchmarks (where the normalized regret for an HPO

Table 6: Consensus Rankings of HPO Methods for Real, Surrogate and Tabular Benchmarks.

| Benchmark | Consensus Ranking (CR) | Permutation Order |
|---|---|---|
| Real | BO_GP_DF > BO_GP_RS > BO_RF_RS > BO_NN_RS > BO_NN_DF > HB > BO_RF_DF > RS | - |
| Surrogate | BO_GP_DF > BO_GP_RS > BO_RF_RS > BO_NN_RS > HB > BO_NN_DF > BO_RF_DF > RS | 2 |
| Tabular | BO_GP_DF > BO_GP_RS > BO_RF_DF > HB > BO_RF_RS > BO_NN_DF > BO_NN_RS > RS | 5 |

method given a cumulative budget is defined as the difference between the value of the best HPC found by any algorithm and the value of the best HPC found by this method, scaled by the range of objective function values as found by any method, see also [60]). Based on the normalized regret, we also computed the mean rank of each HPO method.

Results for the Branin2D, Currin2D and Hartmann3D benchmark functions are given in Figure 8.

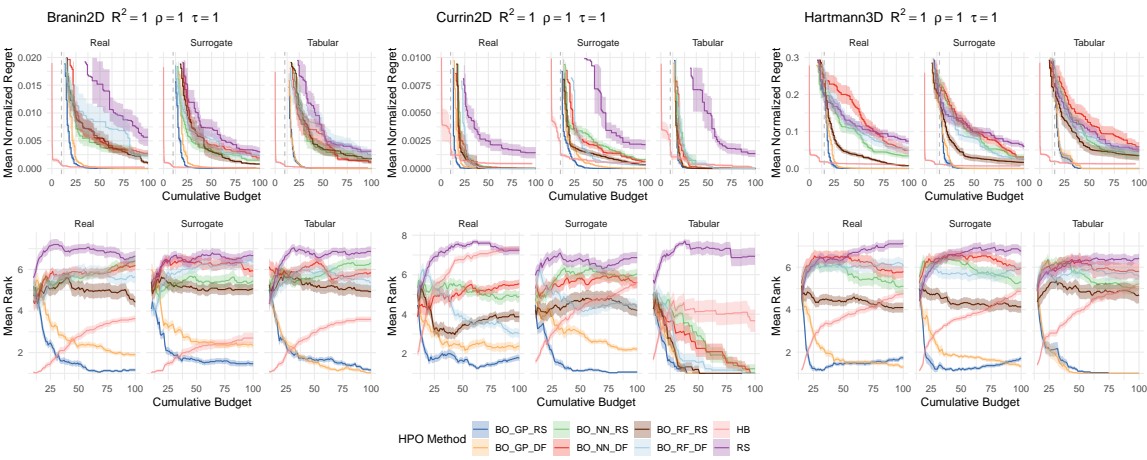

Figure 8: Mean normalized regret (top) and mean ranks (bottom) of different HPO methods on different benchmarks. Ribbons represent standard errors. The gray vertical line indicates the cumulative budget used for the initial design of BO methods. Performance measures of the surrogate benchmarks are stated after the benchmark function. 30 replications.

Differences between tabular and real/surrogate benchmarks can be explained by the fact that the inner optimization problem of BO methods is much easier to solve when only a finite set of potential candidates must be evaluated (i.e., by exhaustive search). We also observe that for the BO performance on the tabular benchmarks, there is no substantial difference in whether the acquisition function optimization is solved exactly or via a Random Search. We employ the rank-based symmetric difference (SD) method which aims to find a consensus ranking that minimizes the average number of rank reversals for the individual benchmark function rankings. We limit ourselves to the scenario of considering the set of all linear orders of HPO methods as candidates for a consensus ranking (SD/L). By comparing the consensus ranking obtained via the surrogate/tabular benchmarks to the consensus ranking obtained using the real benchmarks, we determine the faithfulness of surrogate and tabular benchmarks. We observe that the consensus ranking obtained using the surrogate benchmarks matches the real one more closely than rankings obtained using tabular benchmarks (Table 6).

## E.2 Single-Objective Benchmark on YAHPO-SO

**Instances and Evaluation Protocol.** We use the set of instances and target variables defined for the YAHPO-SO benchmark suite in Supplement C.2 and detailed in Table 3. We furthermore follow the described evaluation protocol, using available search spaces and optimization budgets including 30 replications to assess variance in results. As an evaluation criterion, we report mean normalized

regret (based on the target metric), see Figure 9. Table 7 provides additional info on all optimizers used in the benchmark. Random Search simply samples configurations uniformly at random. SMAC is a model based full-fidelity optimizer using a random forest as surrogate model and Expected Improvement as acquisition function [34]. We use the SMAC4HPO facade [47]. Hyperband randomly samples new configurations and allocates more fidelity to promising configurations by relying on repeated successive halving (SH; [32]). BOHB combines BO with Hyperband and uses a Tree Parzen Estimator (TPE; [5]) as surrogate model. DEHB is a model-free successor of BOHB which relies on differential evolution instead of BO. We use the software defaults regarding the choice of mutation and crossover. SMAC-HB also combines BO with Hyperband but uses a random forest as surrogate model (SMAC4MF facade; [47]). Our optuna optimizer uses a TPE as surrogate model and a median pruner [25] that follows a fixed SH schedule. A configuration is stopped by the pruner if its best intermediate result (at a given fidelity level determined by the SH schedule) is worse compared to the median of the other configurations on the same fidelity level.

Table 7: Optimizers used in the single-objective benchmark.

| Optimizer | Software | Reference | Version |
|---|---|---|---|
| Random Search | - | - | - |
| SMAC (SMAC4HPO) | https://github.com/automl/SMAC3 | [47] | 1.1.1 |
| Hyperband | https://github.com/automl/HpBandSter | [44] | 0.7.4 |
| BOHB | https://github.com/automl/HpBandSter | [21] | 0.7.4 |
| DEHB | https://github.com/automl/DEHB | [4] | 67ac239 |
| SMAC-HB (SMAC4MF) | https://github.com/automl/SMAC3 | [47] | 1.1.1 |
| optuna | https://optuna.org/ | [2] | 2.10.0 |

### E.3 Multi-Objective Benchmark on YAHPO-MO

**Instances and Evaluation Protocol.** We use the set of instances and target variables defined for the YAHPO-MO benchmark suite in Supplement C.2 and detailed in Table 4. We furthermore follow the described evaluation protocol, using available search spaces and optimization budgets including 30 replications to assess variance in results. As an evaluation criterion, we report the mean Hypervolume Indicator [80] computed on normalized targets (see Figure 10). Nadir points and reference Pareto fronts were obtained empirically over all replications of all HPO methods on a given benchmark instance. Table 8 provides additional info on all optimizers used in the benchmark. Random Search simply samples configurations uniformly at random. Random Search (x4) at each step samples four configurations uniformly at random (in parallel). We include this variant as a strong baseline. ParEGO is a model based optimizer relying on a scalarization of the objectives which we then model using a random forest as surrogate model. As acquisition function we use Expected Improvement [34]. SMS-EGO is a model based optimizer that uses a surrogate model for each objective (again, we use random forests) and proposes candidates based on the $\mathcal{S}$-metric [61]. EHVI is a model based optimizer using a surrogate model for each objective (again, we use random forests) and proposes candidates based on their Expected Hypervolume Improvement [20]. MEGO is a model based optimizer using a surrogate model for each objective (again, we use random forests) and proposes candidates by considering the Expected Improvement for each objective which gives rise to a multi-objective optimization problem of the acquisition functions themselves. For the final candidate selection, we sample uniformly at random over the Pareto optimal (with respect to the Expected Improvements) candidates. MIES is a mixed integer evolutionary optimizer (plus survival scheme, $\mu = \lfloor \text{budget}/6 \rfloor$, $\lambda = \lfloor \mu/4 \rfloor$[8]). We use Gaussian mutation ($p = 0.2$)

---

[8]where budget is the optimization budget for a given instance, i.e., number of total evaluations

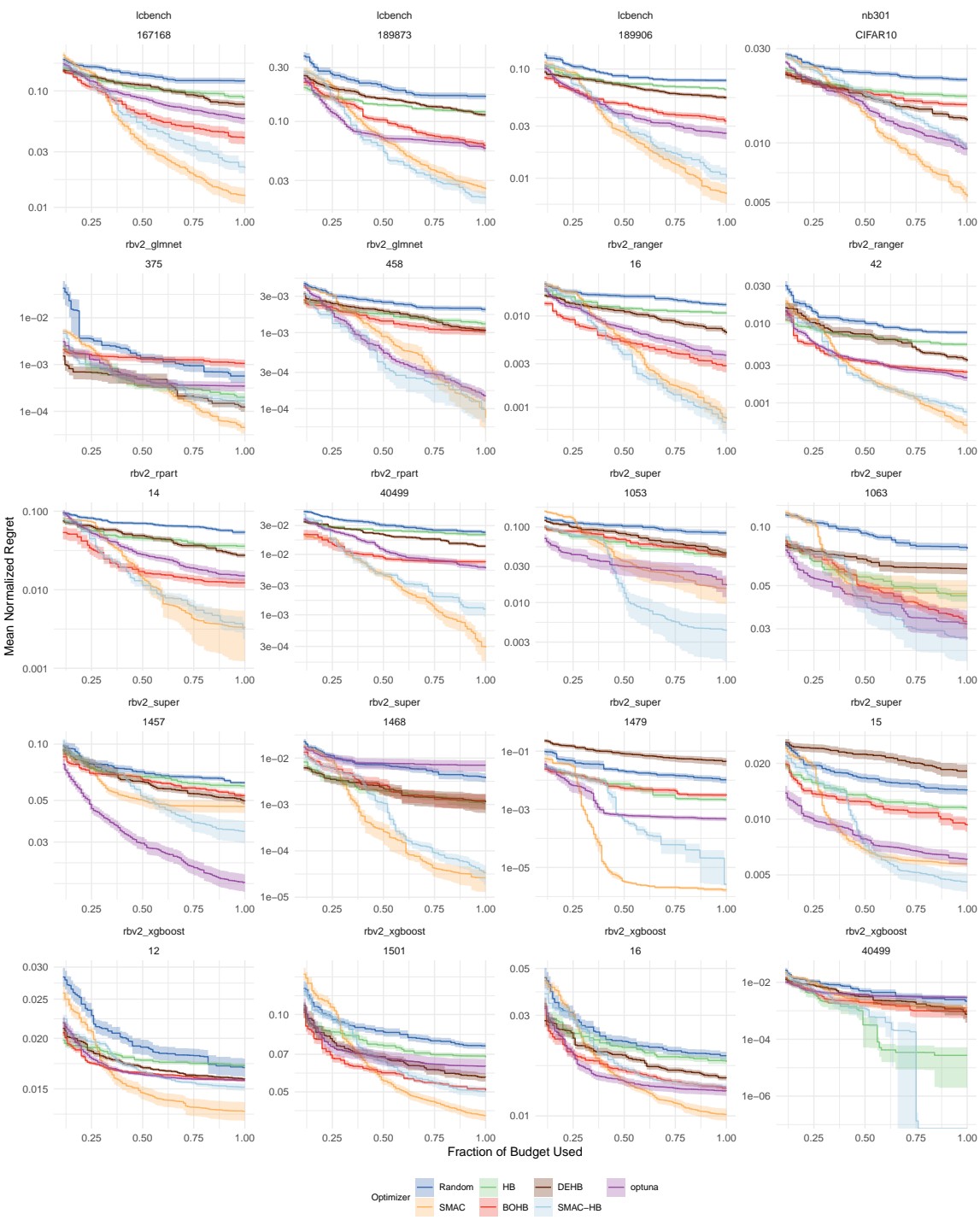

Figure 9: Mean normalized regret of HPO methods separate for each benchmark instance. x-axis starts after 10%.

for numerical parameters and discrete uniform mutation ($p = 0.2$) for categorical parameters. For recombination, we use uniform crossover ($p = 0.2$). As parent selection we perform a tournament selection of parents using nondominated sorting. For survival, we select the best individuals based on nondominated sorting.

Table 8: Optimizers used in the multi-objective benchmark.

| Optimizer | Software | Reference | Version |
|---|---|---|---|
| Random Search | - | - | - |
| Random Search (x4) | - | - | - |
| ParEGO | `https://github.com/mlr-org/mlr3mbo` | [40] | 1f59e13 |
| SMS-EGO | `https://github.com/mlr-org/mlr3mbo` | [61] | 1f59e13 |
| EHVI | `https://github.com/mlr-org/mlr3mbo` | [20] | 1f59e13 |
| MEGO | `https://github.com/mlr-org/mlr3mbo` | [33] | 1f59e13 |
| MIES | `https://github.com/mlr-org/miesmuschel` | [46] | 3483f11 |

## F  Scenarios, Search Spaces and Data Sources

### Random Bot V2 (rbv2_)

All scenarios prefixed with *rbv2_* use data described in [9]. Data contains results from several ML algorithms trained across up to 119 datasets evaluated for a large amount of random evaluations. Table 9 lists all hyperparameters of the search space of the *rbv2_* scenarios. Targets are given by accuracy (acc), balanced accuracy (bac), AUC (auc), Brier Score (`brier`), F1 (`f1`), log loss (`logloss`), time for training the model (`timetrain`), and memory usage (`memory`).
Surrogates are fitted on subsets of the full data available from [9], such that a minimum of 1500 and a maximum of 200000 (depending on the scenario) evaluations are available for each instance in each scenario. All scenarios consist of a pre-processing step (missing data imputation) and a subsequently fitted ML algorithm. Instance ID's correspond to OpenML [71] *dataset* ids through which dataset properties can be queried[9]. OpenML tasks corresponding to each dataset can be obtained from [9]. We abbreviate the *num.impute.selected.cpo* hyperparameter with *imputation* throughout the tables. We fix the *repl* parameter to 10 for experiments.

### NAS-Bench-301 (nb301)

*nb301* uses data of the NAS-Bench-301 benchmark ([78], see also [65]). Table 10 lists all hyperparameters of the search space of the *nb301* scenario. Targets are given by the validation accuracy (`val_accuracy`) and the training time (`runtime`).

### LCBench (lcbench)

The *lcbench* collection uses data of the LCBench benchmark [77], as described in [79]. Table 11 lists all hyperparameters of the search space of the *lcbench* scenario. Targets are given by the validation accuracy (`val_accuracy`), validation cross entropy (`val_crossentropy`), validation balanced accuracy (`val_balanced_accuracy`), test cross entropy (`test_crossentropy`), test balanced accuracy (`test_balanced_accuracy`) and the training time (`time`). Instance ID's correspond to OpenML [71] *task* ids through which task properties can be queried[10] The task with the ID 167083 exhibited unnatural learning curves and was therefore excluded.

---

[9]`https://www.openml.org/d/<dataset_id>`
[10]`https://www.openml.org/t/<task_id>`

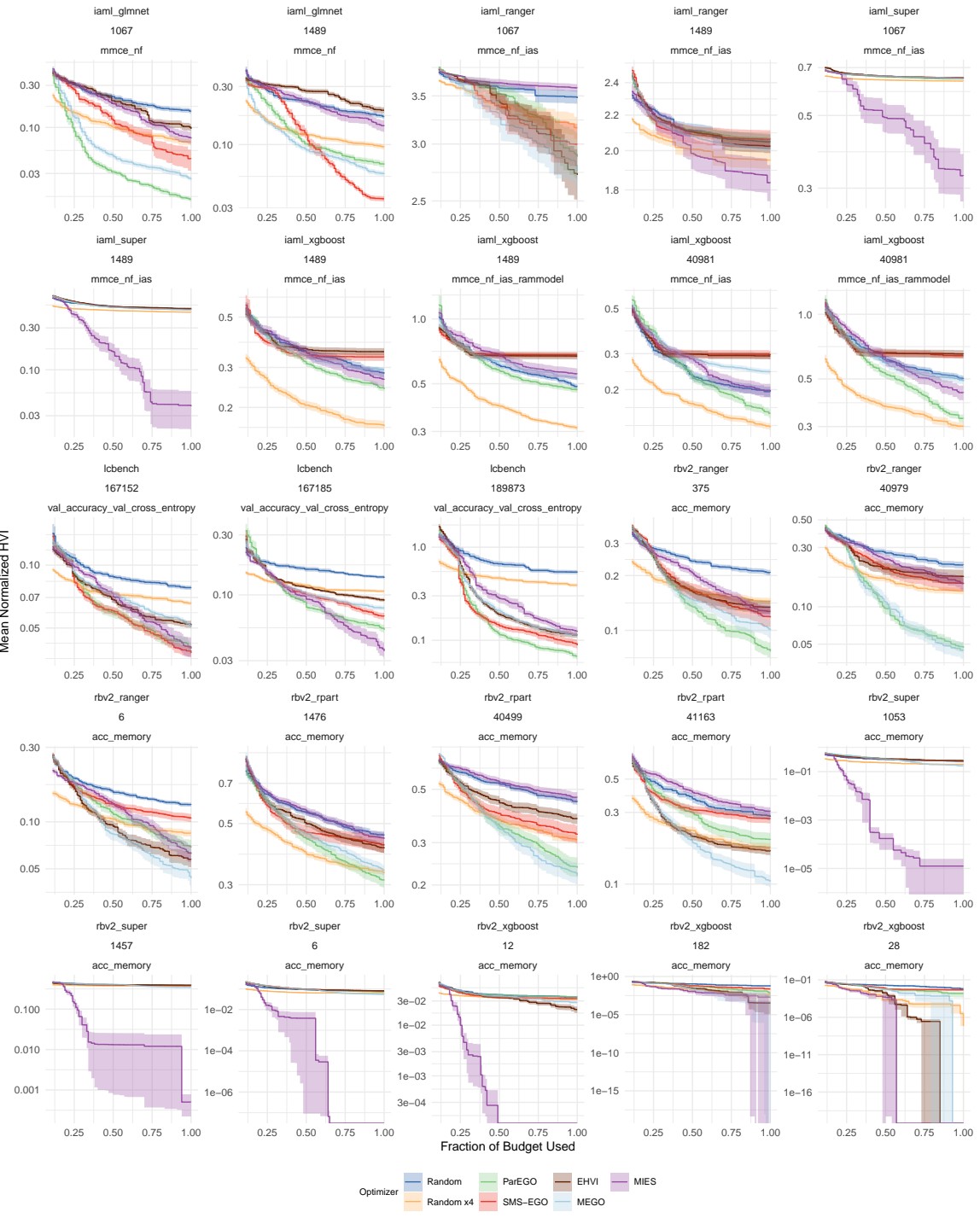

Figure 10: Mean normalized Hypervolume Indicator of HPO methods separate for each benchmark instance. x-axis starts after 10%.

Table 9: Search spaces of YAHPO Gym's *rbv2_* scenarios. ⊢ indicates the parent in case dependencies between hyperparameters exist. The *super* scenario inherits dependencies from previous scenarios, while additional dependencies on the learner_id are introduced, indicated by a prefix.

### *rbv2_glmnet*

| Hyperparameter | Type | Range | Info |
|---|---|---|---|
| alpha | continuous | [0, 1] | |
| s | continuous | [0.001, 1097] | log |
| trainsize | continuous | [0.03, 1] | budget |
| imputation | categorical | impute.{mean, median, hist} | |

### *rbv2_rpart*

| Hyperparameter | Type | Range | Info |
|---|---|---|---|
| cp | continuous | [0.001, 1] | log |
| maxdepth | integer | [1, 30] | |
| minbucket | integer | [1, 100] | |
| minsplit | integer | [1, 100] | |
| trainsize | continuous | [0.03, 1] | budget |
| imputation | categorical | impute.{mean, median, hist} | |

### *rbv2_svm*

| Hyperparameter | Type | Range | Info |
|---|---|---|---|
| kernel | categorical | {linear, polynomial, radial} | |
| cost | continuous | [4.5e-05, 2.2e4] | log |
| gamma | continuous | [4.5e-05, 2.2e4] | log, ⊢ kernel |
| tolerance | continuous | [4.5e-05, 2] | log |
| degree | integer | [2, 5] | ⊢ kernel |
| trainsize | continuous | [0.03, 1] | budget |
| imputation | categorical | impute.{mean, median, hist} | |

### *rbv2_aknn*

| Hyperparameter | Type | Range | Info |
|---|---|---|---|
| k | integer | [1, 50] | |
| distance | categorical | {l2, cosine, ip} | |
| M | integer | [18, 50] | |
| ef | integer | [7, 403] | log |
| ef_construction | integer | [7, 403] | log |
| trainsize | continuous | [0.03, 1] | budget |
| imputation | categorical | impute.{mean, median, hist} | |

### *rbv2_ranger*

| Hyperparameter | Type | Range | Info |
|---|---|---|---|
| num.trees | integer | [1, 2000] | |
| sample.fraction | continuous | [0.1, 1] | |
| mtry.power | integer | [0, 1] | |
| respect.unordered.factors | categorical | {ignore, order, partition} | |
| min.node.size | integer | [1, 100] | |
| splitrule | categorical | {gini, extratrees} | |
| num.random.splits | integer | [1, 100] | ⊢ splitrule |
| trainsize | continuous | [0.03, 1] | budget |
| imputation | categorical | impute.{mean, median, hist} | |

### *rbv2_xgboost*

| Hyperparameter | Type | Range | Info |
|---|---|---|---|
| booster | categorical | {gblinear, gbtree, dart} | |
| nrounds | integer | [7, 2980] | log |
| eta | continuous | [0.001, 1] | log, ⊢ booster |
| gamma | continuous | [4.5e-05, 7.4] | log, ⊢ booster |
| lambda | continuous | [0.001, 1097] | log |
| alpha | continuous | [0.001, 1097] | log |
| subsample | continuous | [0.1, 1] | |
| max_depth | integer | [1, 15] | ⊢ booster |
| min_child_weight | continuous | [2.72, 148.4] | log, ⊢ booster |
| colsample_bytree | continuous | [0.01, 1] | ⊢ booster |
| colsample_bylevel | continuous | [0.01, 1] | ⊢ booster |
| rate_drop | continuous | [0, 1] | ⊢ booster |
| skip_drop | continuous | [0, 1] | ⊢ booster |
| trainsize | continuous | [0.03, 1] | budget |
| imputation | categorical | impute.{mean, median, hist} | |

### *rbv2_super*

| Hyperparameter | Type | Range | Info |
|---|---|---|---|
| svm.kernel | categorical | {linear, polynomial, radial} | |
| svm.cost | continuous | [4.5e-05, 2.2e4] | log |
| svm.gamma | continuous | [4.5e-05, 2.2e4] | log |
| svm.tolerance | continuous | [4.5e-05, 2] | log |
| svm.degree | integer | [2, 5] | |
| glmnet.alpha | continuous | [0, 1] | |
| glmnet.s | continuous | [0.001, 1097] | log |
| rpart.cp | continuous | [0.001, 1] | log |
| rpart.maxdepth | integer | [1, 30] | |
| rpart.minbucket | integer | [1, 100] | |
| rpart.minsplit | integer | [1, 100] | |
| ranger.num.trees | integer | [1, 2000] | |
| ranger.sample.fraction | continuous | [0.1, 1] | |
| ranger.mtry.power | integer | [0, 1] | |
| ranger.respect.unordered.factors | categorical | {ignore, order, partition} | |
| ranger.min.node.size | integer | [1, 100] | |
| ranger.splitrule | categorical | {gini, extratrees} | |
| ranger.num.random.splits | integer | [1, 100] | |
| aknn.k | integer | [1, 50] | |
| aknn.distance | categorical | {l2, cosine, ip} | |
| aknn.M | integer | [18, 50] | |
| aknn.ef | integer | [7, 403] | log |
| aknn.ef_construction | integer | [7, 403] | log |
| xgboost.booster | categorical | {gblinear, gbtree, dart} | |
| xgboost.nrounds | integer | [7, 2980] | log |
| xgboost.eta | continuous | [0.001, 1] | log |
| xgboost.gamma | continuous | [4.5e-05, 7.4] | log |
| xgboost.lambda | continuous | [0.001, 1097] | log |
| xgboost.alpha | continuous | [0.001, 1097] | log |
| xgboost.subsample | continuous | [0.1, 1] | |
| xgboost.max_depth | integer | [1, 15] | |
| xgboost.min_child_weight | continuous | [2.72, 148.41] | log |
| xgboost.colsample_bytree | continuous | [0.01, 1] | |
| xgboost.colsample_bylevel | continuous | [0.01, 1] | |
| xgboost.rate_drop | continuous | [0, 1] | |
| xgboost.skip_drop | continuous | [0, 1] | |
| trainsize | continuous | [0.03, 1] | budget |
| imputation | categorical | impute.{mean, median, hist} | |
| learner_id | categorical | {aknn, glmnet, ranger, rpart, svm, xgboost} | |

Table 10: Search space of the *nb301* scenario. We summarize multiple parameters (using, e.g., {3 − 5} if parameters with suffix 3 through 5 are present).

| Hyperparameter | Type | Range | Info |
|---|---|---|---|
| NetworkSelectorDatasetInfo_COLON_darts_COLON_edge_normal_{0-13} | categorical | {max_pool_3x3, avg_pool_3x3, skip_connect, sep_conv_3x3, sep_conv_5x5, dil_conv_3x3, dil_conv_5x5} | |
| NetworkSelectorDatasetInfo_COLON_darts_COLON_edge_reduce_{0-13} | categorical | {max_pool_3x3, avg_pool_3x3, skip_connect, sep_conv_3x3, sep_conv_5x5, dil_conv_3x3, dil_conv_5x5} | |
| NetworkSelectorDatasetInfo_COLON_darts_COLON_inputs_node_normal_{3-5} | categorical | {0_1, 0_2, 1_2} | |
| NetworkSelectorDatasetInfo_COLON_darts_COLON_inputs_node_reduce_{3-5} | categorical | {0_1, 0_2, 1_2} | |
| epoch | integer | [1, 98] | budget |

Table 11: Search space of the *lcbench* scenario.

| Hyperparameter | Type | Range | Info |
|---|---|---|---|
| epoch | integer | [1, 52] | budget |
| batch_size | integer | [16, 512] | log |
| learning_rate | continuous | [1e-04, 0.1] | log |
| momentum | continuous | [0.1, 0.9] | |
| weight_decay | continuous | [1e-05, 0.1] | |
| num_layers | integer | [1, 5] | |
| max_units | integer | [64, 1024] | log |
| max_dropout | continuous | [0, 1] | |

**Interpretable AutoML (iaml_)**

All scenarios prefixed with *iaml_* rely on data that were newly collected by us. Different `mlr3` [41] learners ("classif.glmnet", "classif.rpart", "classif.ranger", "classif.xgboost") were incorporated into an ML pipeline with minimal preprocessing (removing constant features, fixing unseen factor levels during prediction and missing value imputation for factor variables by sampling from non-missing training levels) via `mlr3pipelines` [8]. Hyperparameters of the learners were sampled uniformly at random (for the search spaces, see Table 12) and the ML pipeline performance (classification error - mmce, F1 score - f1, AUC - auc, logloss - `logloss`) was evaluated via 5-fold cross-validation on the following OpenML [71] datasets (dataset id): 40981, 41146, 1489, 1067. Each pipeline was then refitted and used for prediction on the whole data to estimate training and predict time (`timetrain`, `timepredict`) and RAM usage (during training and prediction, `ramtrain` and `rampredict` as well as model size, `rammodel`). Moreover, interpretability measures as described in [52] were computed for all models: number of features used (nf), interaction strength of features (ias) and main effect complexity of features (mec). To our best knowledge, this is the first publicly available benchmark that combines performance, resource usage and interpretability of models allowing for the construction of interesting multi-objective benchmarks. Hyperparameter configurations were evaluated at different fidelity steps (training sizes of the following fractions: 0.05, 0.1, 0.2, 0.4, 0.6, 0.8, 1) achieved via incorporating resampling in the ML pipeline. The super learner scenario was constructed by using the data of all four base learners introducing conditional hyperparameters in the form of branching. In total, 5451872 different configurations were evaluated. Data collection was performed on the *moran* partition of the *ARCC Teton HPC* cluster of the University of Wyoming using `batchtools` [42] for job scheduling and took around 9.8 CPU years. Surrogate models were then fitted on the available data as described in Supplement D.1. Table 12 lists all hyperparameters of the search spaces of the *iaml_* scenarios. Instance ID's correspond to OpenML [71] *dataset* ids through which dataset properties can be queried[11].

---

[11]https://www.openml.org/d/<dataset_id>

Table 12: Search spaces of YAHPO Gym's *iaml_* scenarios. ⊢ indicates the parent in case dependencies between hyperparameters exist. The *super* scenario inherits dependencies from previous scenarios, while additional dependencies on the learner are introduced, indicated by a prefix.

### *iaml_glmnet*

| Hyperparameter | Type | Range | Info |
|---|---|---|---|
| alpha | continuous | [0, 1] | |
| s | continuous | [1e-04, 1000] | log |
| trainsize | continuous | [0.03, 1] | budget |

### *iaml_rpart*

| Hyperparameter | Range | Type | Info |
|---|---|---|---|
| cp | continuous | [1e-04, 1] | log |
| maxdepth | integer | [1, 30] | |
| minbucket | integer | [1, 100] | |
| minsplit | integer | [1, 100] | |
| trainsize | continuous | [0.03, 1] | budget |

### *iaml_ranger*

| Hyperparameter | Type | Range | Info |
|---|---|---|---|
| num.trees | integer | [1, 2000] | |
| replace | boolean | {TRUE, FALSE} | |
| sample.fraction | continuous | [0.1, 1] | |
| mtry.ratio | continuous | [0, 1] | |
| respect.unordered.factors | categorical | {ignore, order, partition} | |
| min.node.size | integer | [1, 100] | |
| splitrule | categorical | {gini, extratrees} | |
| num.random.splits | integer | [1, 100] | ⊢ splitrule |
| trainsize | continuous | [0.03, 1] | budget |

### *iaml_xgboost*

| Hyperparameter | Type | Range | Info |
|---|---|---|---|
| booster | categorical | {gblinear, gbtree, dart} | |
| nrounds | integer | [3, 2000] | log |
| eta | continuous | [1e-04, 1] | log, ⊢ booster |
| gamma | continuous | [1e-04, 7] | log, ⊢ booster |
| lambda | continuous | [1e-04, 1000] | log |
| alpha | continuous | [1e-04, 1000] | log |
| subsample | continuous | [0.1, 1] | |
| max_depth | integer | [1, 15] | ⊢ booster |
| min_child_weight | continuous | [exp(1), 150] | log, ⊢ booster |
| colsample_bytree | continuous | [0.01, 1] | ⊢ booster |
| colsample_bylevel | continuous | [0.01, 1] | ⊢ booster |
| rate_drop | continuous | [0, 1] | ⊢ booster |
| skip_drop | continuous | [0, 1] | ⊢ booster |
| trainsize | continuous | [0.03, 1] | budget |

### *iaml_super*

| Hyperparameter | Type | Range | Info |
|---|---|---|---|
| learner | categorical | {glmnet, rpart, ranger, xgboost} | |
| glmnet.alpha | continuous | [0, 1] | |
| glmnet.s | continuous | [1e-04, 1000] | log |
| rpart.cp | continuous | [1e-04, 1] | log |
| rpart.maxdepth | integer | [1, 30] | |
| rpart.minbucket | integer | [1, 100] | |
| rpart.minsplit | integer | [1, 100] | |
| ranger.num.trees | integer | [1, 2000] | |
| ranger.replace | boolean | {TRUE, FALSE} | |
| ranger.sample.fraction | continuous | [0.1, 1] | |
| ranger.mtry.ratio | continuous | [0, 1] | |
| ranger.respect.unordered.factors | categorical | {ignore, order, partition} | |
| ranger.min.node.size | integer | [1, 100] | |
| ranger.splitrule | categorical | {gini, extratrees} | |
| ranger.num.random.splits | integer | [1, 100] | |
| xgboost.booster | categorical | {gblinear, gbtree, dart} | |
| xgboost.nrounds | integer | [3, 2000] | log |
| xgboost.eta | continuous | [1e-04, 1] | log |
| xgboost.gamma | continuous | [1e-04, 7] | log |
| xgboost.lambda | continuous | [1e-04, 1000] | log |
| xgboost.alpha | continuous | [1e-04, 1000] | log |
| xgboost.subsample | continuous | [0.1, 1] | |
| xgboost.max_depth | integer | [1, 15] | |
| xgboost.min_child_weight | continuous | [exp(1), 150] | log |
| xgboost.colsample_bytree | continuous | [0.01, 1] | |
| xgboost.colsample_bylevel | continuous | [0.01, 1] | |
| xgboost.rate_drop | continuous | [0, 1] | |
| xgboost.skip_drop | continuous | [0, 1] | |
| trainsize | continuous | [0.03, 1] | budget |

