# OpenReview forum: "YAHPO Gym - An Efficient Multi-Objective Multi-Fidelity Benchmark for Hyperparameter Optimization"
_automl.cc/AutoML/2022/Track/Main — AutoML-Conf 2022 (Main Track)_

### Official Review · Reviewer_PsaJ · 2022-03-27

**Potential Impact On The Field Of Automl Rating:** 4
**Technical Quality And Correctness:** It is correct and well-established.
**Technical Quality And Correctness Rating:** 3
**Clarity:** It is clear.
**Clarity Rating:** 4

**Summary Of Contributions:**

This paper describes an efficient multi-objective multi-fidelity benchmark framework for hyperparameter optimization, which is referred to as YAHPO Gym.  In particular, YAHPO Gym is a surrogate-based benchmark for hyperparameter optimization.  Since a tabular benchmark may reduce a bias in performance estimation and ranking of hyperparameter optimization methods.  Moreover, this YAHPO Gym allows us to use multi-objective and multi-fidelity settings.  Finally, the authors demonstrate the comparisons between YAHPO Gym and other available implementations.

**Ethics Details (Optional):**

I think that this paper does not have any ethical issues.

**Overall Review:**

I agree with the motivation of this paper.  It provides an interesting benchmark, in order to answer to the motivation.  It will be a potentially impactful project.

I have a question on the multi-fidelity setting.  How did you control the degree of fidelities?  It is not described in the paper.  I guess you did control the size of training data or any others?

Also, it is a minor issue, but could you provide how to pronounce YAHPO?

**Potential Impact On The Field Of Automl:**

It has a big potential impact on the field of AutoML, because a general framework for mutli-objective and mutli-fidelity hyperparameter optimization (or Bayesian optimization) is currently not available.  This framework will encourage the AutoML community to develop more sophisticated studies and implementations.

**Reproducibility:**

This paper is reproducible and the YAPHO Gym project is also open-source.

**Review Confidence:**

4: You are confident in your assessment, but not absolutely certain. It is unlikely, but not impossible, that you did not understand some parts of the submission or that you are unfamiliar with some pieces of related work.

**Review Rating:**

5: Accept, good paper

**Review Summary:**

Since this paper provides a solid benchmark for hyperparameter optimization including the settings with multiple objectives and multiple fidelities, it is novel and interesting.  The authors describe why we need this benchmark and how it will be used thoroughly.  Therefore, I would like to recommend the acceptance of this work.

---

### Official Review · Reviewer_PonF · 2022-04-02

**Potential Impact On The Field Of Automl:** N/A for reproducibility reviewers
**Potential Impact On The Field Of Automl Rating:** 3
**Technical Quality And Correctness:** N/A for reproducibility reviewers
**Technical Quality And Correctness Rating:** 4
**Clarity:** N/A for reproducibility reviewers
**Clarity Rating:** 4

**Summary Of Contributions:**

N/A for reproducibility reviewers

**Overall Review:**

YAHPO_GYM is clearly presented with every small detail is nicely covered. The reproducibility list is filled out fully, the answers are clearly justified. The results are adequately provided. Installation is a slightly challenging task as it requires cloning the second github repo. However, it is totally justifiable. Following the instructions provided by the authors, the user can easily reproduce the examples. Integrating YAHPO_GYM in other algorithms should be straightforward. For example, the example with BOHB provides clear instructions. In general, one needs to be familiar with ConfigSpace. Overall, it is easy to use YAHPO_GYM.

**Reproducibility:**

YAHPO_GYM is clearly presented with every small detail is nicely covered. The reproducibility list is filled out fully, the answers are clearly justified. The results are adequately provided. Installation is a slightly challenging task as it requires cloning the second github repo. However, it is totally justifiable. Following the instructions provided by the authors, the user can easily reproduce the examples. Integrating YAHPO_GYM in other algorithms should be straightforward. For example, the example with BOHB provides clear instructions. In general, one needs to be familiar with ConfigSpace. Overall, it is easy to use YAHPO_GYM.

**Review Confidence:**

5: You are absolutely certain about your assessment. You are very familiar with the related work and checked all the details carefully.

**Review Rating:**

5: Accept, good paper

**Review Summary:**

N/A for reproducibility reviewers

---

### Official Review · Reviewer_Kp9M · 2022-04-03

**Potential Impact On The Field Of Automl Rating:** 4
**Technical Quality And Correctness Rating:** 4
**Clarity Rating:** 4

**Summary Of Contributions:**

This paper presents the YAHPO gym system as a benchmarking tool for multi-objective and multi-fidelity hyperparameter optimization. This benchmark makes use of surrogate functions instead of real evaluations or tabular benchmarks as a compromise between the computational cost of real evaluations and the bias introduced by the tabular benchmarks that rely on explicit discretization. This benchmarking tool contains problems with a wide range of search spaces aand also provides the ability to evaluate transfer HPO algorithms. One significant feature of this benchmark (over existing benchmarks) is the efficiency in terms of time and memory overhead which is obtained via the use of ResNet based compressed neural network surrogates. This allows for a faithful evaluation of various forms of hyperparameter optimization algorithms with reduced overall computational cost, allowing for efficient evaluation between algorithms with multiple repetitions, making it easy to get reliable statistically significant comparisons.


**Clarity:**

The paper is very clear in its presentation and various aspects are clearly documented in the supplement.


**Overall Review:**

Positive aspects:
- The proposed system contains various scenarios that allow one to evaluate various aspects of any HPO algorithm efficiently with low memory overhead (compared to existing benchmarks).
- The paper is very well written and easy to follow, properly motivating and describing the different choices for the system and justifying them empirically and providing sufficient amount of details in the supplement.


Negative aspect:
- In my opinion, there are no negative aspects to this paper.


**Potential Impact On The Field Of Automl:**

Efficient established benchmarks are critical for any field of research. Making a benchmark faithful yet more efficient makes it easy for algorithm developers to perform a statistically significant evaluation. If the benchmarks take too much time or computational resources, it creates a barrier for the algorithm developers. Having a benchmark which requires an order of magnitude less time and memory lowers this evaluation barrier significantly, and brings us closer to the point in the area of HPO research where there is very little excuse of HPO algorithm developers to not evaluate on such benchmarks.

This proposed benchmarking system also handles various aspects of the ML HPO problem landscape such as multi-fidelity, multi-objective, as well as hierarchical hyperparameter search spaces, allowing one to evaluate HPO algorithm across various dimensions.


**Reproducibility:**

The paper provides enough details in terms of the checklist and the system that makes it straightforward for anyone to replicate the results.


**Review Confidence:**

3: You are fairly confident in your assessment. It is possible that you did not understand some parts of the submission or that you are unfamiliar with some pieces of related work.

**Review Rating:**

6: Strong accept, should be highlighted

**Review Summary:**

This is a very strong HPO benchmarking system covering all relevant aspects of the HPO problem, and provides a more faithful yet more efficient (in terms of time and memory overhead) comparison between HPO algorithms. For this reason, I strongly recommend the acceptance of this paper.


**Technical Quality And Correctness:**

In terms of technical quality, the paper empirically justifies the choice of surrogate functions over tabular benchmarks for a representative set of HPO algorithms. The results indicate that the surrogate function based benchmarks are more faithful to the real evaluation based benchmarks compared to the tabular benchmarks.

Given this benchmark, the paper studies two questions RQ1 and RQ2. There are couple of questions I have regarding these research questions. First, it is not clearly motivated as to why these research questions are selected. The answers to these questions are usually well accepted in literature (and the answers found in this paper corroborate that). Second, it is not clear how different the answer would be if we used some of the other benchmarks (such as HPOBench and HPO-B) with some level of parity (for example, controlling the number of repetitions such that all benchmarks take the same amount of time). If the claim is that YAHPO should be used in place of these existing benchmarks, it would be good to understand (i) which benchmark provides the most faithful comparison (when compared to real evaluations), and (ii) what are the computational costs for the evaluation of each of the benchmarks. Alternately, it would be good to understand why such an evaluation between benchmarks is not meaningful.

Note that the above concern is somewhat limited since the proposed system covers problem classes (multi-objective, multi-fidelity, hierarchical search spaces) that existing benchmarks do not cover.

---

### Official Review · Reviewer_j3GQ · 2022-04-04

**Potential Impact On The Field Of Automl Rating:** 3
**Technical Quality And Correctness Rating:** 4
**Clarity Rating:** 4

**Summary Of Contributions:**

This paper proposes a benchmark tool for hyper-parameter Bayesian optimization. It includes many problems where one can test a Bayesian optimization method. Including multi-fidelity and multi-objective problems. It is based on using a surrogate model to predict the unknown target function. This model is a neural network that has been trained with a large number of instances of the original problem. The framework is fast and memory efficient.


**Clarity:**

The paper is correctly written and specific details are given.

The paper contains missing references that appear as ??.


**Overall Review:**

Overall I think that this is an interesting paper that proposes a tool that the community will most likely find useful.


**Potential Impact On The Field Of Automl:**

I think this work is likely to receive the attention of the community. It provides a mechanism to easily compare BO methods on a wide range of problems very easily. It also has improved features with respect to already existing alternatives.


**Reproducibility:**

The source code of the benchmark framework is given. This guarantees reproducibility.


**Review Confidence:**

3: You are fairly confident in your assessment. It is possible that you did not understand some parts of the submission or that you are unfamiliar with some pieces of related work.

**Review Rating:**

5: Accept, good paper

**Review Summary:**

A nice paper providing a useful tool for benchmarking BO methods in the task of finding good hyper-parameters. The utility of the framework is illustrated with several experiments.

**Technical Quality And Correctness:**

The writing of the paper seems correct and proper references are cited. The paper also evaluates the utility of using surrogate model showing that it gives results that are closer to the actual ones. The benchmark framework is used to evaluate several BO methods showing results that seem reasonable and that do correspond to the literature.

---

### Official Review · Reviewer_9dTp · 2022-04-06

**Potential Impact On The Field Of Automl:** A benchmark for multi-fidelity multi-…
**Potential Impact On The Field Of Automl Rating:** 2
**Technical Quality And Correctness:** Technically sound.
**Technical Quality And Correctness Rating:** 3
**Clarity Rating:** 1

**Summary Of Contributions:**

The paper proposes YAHPO-Gym, a surrogate-based benchmark for (multi-objective) multi-fidelity hyperparameter optimization problems with 14 configuration spaces consisting of multiple instances to facilitate transfer learning. The authors empirically address the performance of several multi-fidelity approaches compared to random search, as well as the quality of advanced multi-fidelity methods.

**Clarity:**

The paper is not very clear. Namely, there are alot of missing references, the authors rely on their publicly available work, for most of the information. The paper is confusing in its current state.

**Overall Review:**

Check review summary.

**Reproducibility:**

The repository is accessible.

**Review Confidence:**

5: You are absolutely certain about your assessment. You are very familiar with the related work and checked all the details carefully.

**Review Rating:**

2: Reject, not good enough

**Review Summary:**

The paper proposes YAHPO-Gym, a surrogate-based benchmark for (multi-objective) multi-fidelity hyperparameter optimization problems with 14 configuration spaces consisting of multiple instances to facilitate transfer learning. The authors empirically address the performance of several multi-fidelity approaches compared to random search, as well as the quality of advanced multi-fidelity methods.

**Strengths**

- provides an easy-to-use GYM environment with 14 benchmarks for multi-objective multi-fidelity HPO.

- The authors seized the opportunity to benchmark some multi-objective Bayesian optimization solutions as well as multi-fidelity methods.


**Weaknesses**

- The diversity of the instances for each benchmark collection is not studied. One suggestion would be to visualize the empirical cumulative distribution of the underlying tabular datasets.

- I would be interested to hear more about why the authors chose a deterministic surrogate benchmark instead of a surrogate that provides some uncertainty quantification.

- YAHPO-GYM does not contribute any new benchmarks, but rather provides a wrapper for existing ones.

I think the paper has slightly improved  over the previous submissions, especially with regards to benchmarking the baselines. However, the paper seems to be a not complete with several missing references that makes it difficult to follow through in a self-contained fashion.

---

### Official Review · Reviewer_Rr9U · 2022-04-07

**Potential Impact On The Field Of Automl Rating:** 2
**Technical Quality And Correctness Rating:** 3
**Clarity Rating:** 3

**Summary Of Contributions:**

The authors propose a new collection of benchmarks for HPO containing 14 scenarios (700 instances). The key focus is on multi-fidelity problems and hierarchical search spaces. Although it is not clear how many of these problems exhibit these features. The library also contains multi-objective problems (which is deemed important by the authors).

The benchmarks are surrogate-based, provided using the ONNX format, and have low memory requirements.

**Clarity:**

In general, the paper is clear and well written. However, something went wrong with the references which are all cited as '??' (both papers, sections, etc.

There are some typos such as:
"BO is configured with algorithm surrogate model either a"

**Overall Review:**

While properties of the scenarios are hinted at in some places, there is no qualitative discussion of the different scenarios and instances. The benchmarks are listed mainly, e.g., in a table in a quantitative way. A large number of benchmark problems (and libraries) is not always better, as it is easy for important conclusions to be hidden among many results. For that reason, it's important for benchmark libraries to give more details about the problems it includes.

Readers would benefit from discussing (a subset of) the different benchmarks in terms of difficulty: complexity of the search space, non-stationarity, what type of algorithms are expected to do well, etc.

For surrogate-based benchmarks the ability to capture the fine detail and stochasticity of the problem is crucial. Figure 2 provides insights into how the surrogate performs, but it could use a more thorough validation of the surrogate itself. In the text, tests for Branin2D, Hartmann3D, and Currin2D are announced, but I could not find any results in the paper (in the figure or tables).

**Potential Impact On The Field Of Automl:**

There already exist quite some benchmarks for HPO. Although there are some merits of YAHPO-GYM (fast, hierarchical search space) I consider the impact average at best.

I would have preferred to have extended one of the existing benchmark collections with the missing (surrogate-based, hierarchical, ...) HPO problems, e.g., COCO or HPOlib/HPOBench, so a common interface is provided to test optimizers. Integration in HPOlib is considered future work.

**Reproducibility:**

I was not able to access https://slds-lmu.github.io/yahpo_exps to reproduce the results.

Although the link to the library itself works, and the raw data used to construct the surrogates is available.

**Review Confidence:**

3: You are fairly confident in your assessment. It is possible that you did not understand some parts of the submission or that you are unfamiliar with some pieces of related work.

**Review Rating:**

5: Accept, good paper

**Review Summary:**

I consider the efficient memory requirements and runtime execution of the benchmarks to be the biggest advantage. While there are many benchmark libraries, having a fast surrogate-based one is useful for the community. Although I'd prefer to extend existing frameworks I can see YAHPO-GYM being useful. However, I could not run the experiments of the paper (the link is not working). Together with the missing references (??) and benchmarks (Branin, Hartman3D,...), it is hard to recommend the paper for acceptance

**Technical Quality And Correctness:**

As far as I can see the results are technically sound, but I was not able to run the experiments myself for further analysis.

---

### Meta-Review · Area_Chair_paYy · 2022-05-07

**Recommendation:** Accept
**Confidence:** 4

**Metareview:**

This work proposed a new HPO surrogate benchmark for 14 scenarios and filled the gap of benchmarks in the multi-objective and multi-fidelity setting. The paper is clearly written and conducted experiments to demonstrate its usefulness. It also provided analysis on several interesting questions such as tabular vs surrogate; multi-fidelity vs full-fidelity. It is open sourced and being integrated into a larger benchmarking framework (HPOBench). I think it will make a good contribution to the HPO community.

---

### Decision · Program_Chairs · 2022-05-13

Accept